# Conversational Markov Chains: A Framework for Behavioral Analysis of Large Language Models

## Abstract

How do you compare two language models that score identically on benchmarks but behave very differently in conversation? One might explore diverse topics fluidly; the other might repeat familiar patterns.

We propose modeling multi-turn conversations as Markov chains over semantic states. By embedding conversation turns and clustering them into discrete states, we extract three interpretable metrics from the transition structure that characterize *behavioral diversity* (how varied responses are), *responsiveness* (how readily conversations shift between topics), and *long-term patterns* (which conversational behaviors dominate over time).

Comparing Llama 3.1 8B and Mistral 7B on 300,000+ conversation turns, we find distinct behavioral signatures: Llama produces more diverse responses and transitions fluidly between topics; Mistral is more predictable and tends to dwell in familiar patterns.

The approach requires substantial data, making it a targeted diagnostic tool rather than a lightweight metric. For practitioners who need to understand *how* models behave—not just what they get right—this framework offers a principled, quantitative approach.

## 1 Introduction

Consider two language models that achieve similar scores on a reasoning benchmark. One generates highly variable explanations, exploring diverse reasoning pathways across trials. The other produces near-identical explanations, converging to a narrow set of templates. Standard accuracy metrics conflate these distinct behavioral profiles, yet the choice between them could determine whether a model suits exploratory tutoring or standardized documentation.

This paper addresses a gap in LLM evaluation: we lack principled methods for characterizing *how* models behave in conversation, beyond measuring *what* they know. Existing evaluation paradigms—benchmark accuracy (Hendrycks et al., 2020; Liang et al., 2022), judge-based assessment (Zheng et al., 2023)—measure task performance but fail to quantify behavioral properties such as response predictability, semantic exploration patterns, and conversational persistence. As LLMs enter production in critical domains including medical diagnosis (Singhal et al., 2023), legal reasoning (Blair-Stanek et al., 2023), and educational tutoring (Kasneci et al., 2023), the need for rigorous behavioral characterization intensifies.

### 1.1 The Core Insight

Our key insight is that multi-turn conversations can be viewed as trajectories through a space of "conversational situations." When a student asks about atmospheric pressure, then follows up about humidity, then circles back to temperature—the conversation moves through different types of exchanges. These movements form patterns: some models explore broadly, others dwell in familiar territory; some transition fluidly between topics, others persist in the same semantic region.

Markov chains provide a natural mathematical framework for analyzing such patterns. A Markov chain describes how a system moves between discrete states over time, with transitions depending only on the

current state. If we can map conversations to discrete semantic states, we can study how models move between them—and extract meaningful behavioral descriptors from the resulting transition structure.

This perspective differs fundamentally from prior work that models LLMs as Markov chains at the *token* level (Zekri et al., 2024). Token-level models study the architecture's internal dynamics, treating each generated token as a state. In contrast, we model the *conversation itself* as a Markov chain over *semantic states*—clusters of semantically similar conversational situations. Our object of study is not the generation mechanism but the behavioral dynamics of multi-turn interaction.

## 1.2 Our Approach

Before introducing formal machinery, we outline our approach:

1. **Pick a conversational regime.** We focus on a specific type of conversation—in our case study, teacher-student dialogues about climate and desert formation. This domain combines technical content with conversational variety.

2. **Generate many conversations under consistent conditions.** We fix the model, temperature, and prompting style, then generate thousands of conversations with diverse student questions. This samples how the model behaves within the chosen regime.

3. **Turn conversation turns into semantic states.** Each prompt-response pair is embedded into a vector space (using a text embedding model), then clustered into a finite number of discrete states. Each state corresponds to a "type of situation"—for example, "student asking factual follow-up" or "teacher explaining a mechanism."

4. **Track how conversations move between states.** Given many conversations, we observe how they transition from one state to another, building an empirical transition matrix that captures the probability of each possible state-to-state movement.

5. **Check for Markov-like behavior.** Under appropriate choices of clustering granularity and sufficient data, these transitions exhibit approximate time-homogeneity and depend primarily on the current state rather than detailed history. The resulting chains also exhibit irreducibility—all states communicate with each other through some sequence of transitions—which ensures well-defined long-term behavior and stationary distributions.

6. **Extract behavioral observables from the chain.** Once we have a Markov chain, classical theory provides interpretable observables: entropy rate (how unpredictable transitions are), spectral gap (how quickly the chain "forgets" where it started), and stationary distribution (which states the chain preferentially occupies).

Each subsequent section formalizes one of these steps, but the core logic remains: map conversations to states, study the transitions, and interpret the resulting Markov structure as a behavioral signature.

## 1.3 Research Questions

This work addresses two central questions:

**Q1:** *Can we construct stable Markov models of conversational behavior from realistic data volumes?* (Feasibility: sample complexity, convergence, irreducibility)

**Q2:** *Do Markov observables distinguish between models with similar benchmark performance?* (Discriminative power: Llama vs Mistral comparison)

Sections 6–7 address these questions with empirical evidence.

### 1.4 Contributions

This work makes three primary contributions:

1. **A framework for conversational Markov analysis.** We provide a methodology for modeling LLM conversations as Markov chains over semantic states, from data collection through behavioral interpretation.

2. **Three interpretable behavioral observables.** We demonstrate that entropy rate, spectral gap, and stationary distribution provide complementary perspectives on conversational dynamics:
   - Entropy rate quantifies response diversity ("how predictable is the next move?")
   - Spectral gap measures mixing time ("how fast does the conversation forget where it started?")
   - Stationary distribution reveals repertoire breadth ("which semantic regions does the model preferentially occupy?")

3. **Empirical validation through a comparative case study.** We apply the framework to compare Llama 3.1 8B and Mistral 7B on teacher-student dialogues under controlled stateless generation, demonstrating that the observables capture meaningful behavioral differences between models with comparable benchmark performance. The framework complements existing evaluation methods as one tool in a broader measurement toolkit.

### 1.5 Limitations

We emphasize several fundamental constraints that bound the applicability of this methodology:

**Sample complexity.** Reliable Markov chain estimates require substantial data. Our case study achieves $m = 25$ states with $N \approx 100{,}000$–$200{,}000$ transitions (details in Section 3.5). This positions our method as a targeted diagnostic tool suitable for applications willing to invest in large-scale data collection, not a lightweight evaluation alternative.

**Embedding and clustering dependence.** Our analysis views LLM behavior through the lens of a specific embedding space and clustering granularity. Different embeddings or cluster counts may yield different state spaces and thus different observables. The framework measures behavior *relative to these choices*, not in absolute terms.

**Domain specificity.** Our experimental validation focuses on one conversational domain: teacher-student dialogues about atmospheric science. Behavioral signatures may differ across domains (code generation, creative writing, factual Q&A). The specific findings about Llama and Mistral should not be over-interpreted as universal characterizations—they describe behavior within this particular regime.

**Framework contribution vs. specific claims.** The primary contribution is the framework itself—a methodology for behavioral characterization. The specific finding that "Llama is more exploratory than Mistral in teacher-student dialogues" validates the approach but generalizing requires replication across diverse tasks.

## 2 Background and Related Work

Our framework draws on three areas of prior work: Markov models in language processing, LLM evaluation methods, and embedding-based text analysis. We position our contribution at the intersection of these areas, highlighting how conversational-level Markov analysis differs from existing approaches.

## 2.1 LLMs as Markov Chains: Token-Level Approaches

Inspiring recent work has demonstrated the power of viewing LLMs through a Markov chain lens. Zekri et al. (2024) formally model autoregressive LLMs as Markov chains over the token/sequence state space, deriving mixing time bounds and analyzing how temperature affects convergence. This elegant formalization reveals deep connections between language model behavior and classical stochastic process theory, demonstrating that Markov chain concepts—entropy, mixing times, stationary distributions—provide meaningful characterizations of LLM dynamics.

While Markov analysis has provided insights at the token level, its application to conversational-level behavior remains unexplored. We pursue this complementary direction, modeling multi-turn conversations as Markov chains over semantic states rather than token sequences. While the token-level approach illuminates the generation mechanism, our conversational approach illuminates behavioral patterns across extended interactions.

The two perspectives share conceptual tools (entropy, mixing time, stationary distributions) but apply them at distinct granularities:

- **Token-level models** reveal properties of the autoregressive generation process itself.

- **Our conversational model** reveals how those generation properties manifest as observable behavioral trajectories.

Importantly, modeling prompt-response dynamics required turn-level definitions (Appendix A), since the state space and transition semantics differ fundamentally from token-level analysis. The token-level work nevertheless demonstrated that Markov chain analysis could yield interpretable, actionable insights about LLM behavior—motivating our extension to the conversational setting.

## 2.2 LLM Evaluation: Accuracy vs. Behavior

The dominant paradigm in LLM evaluation focuses on task accuracy. Benchmarks such as MMLU (Hendrycks et al., 2020) measure factual knowledge across domains; HELM (Liang et al., 2022) provides holistic assessment across multiple axes; MT-Bench (Zheng et al., 2023) evaluates multi-turn conversation quality through judge-based scoring. These methods answer the question: *what does the model know, and how well does it perform on specific tasks?*

Our framework addresses a different question: *how does the model behave across conversations?* Two models with identical benchmark accuracy may exhibit qualitatively different behavioral dynamics:

- One may generate diverse responses to similar prompts; the other may converge to stereotyped templates.

- One may transition fluidly between topics; the other may persist in the same semantic region.

- One may distribute attention evenly across conversational territory; the other may concentrate in attractor states.

These behavioral properties are invisible to accuracy metrics but may critically affect suitability for different applications. A medical documentation system benefits from predictable, template-adherent responses; a creative writing assistant benefits from diverse, exploratory behavior. Our framework provides quantitative handles on such distinctions.

**Behavioral evaluation beyond accuracy.** Prior work has recognized the limitations of accuracy-focused evaluation. Ribeiro et al. (2020) introduced behavioral testing through CheckList, probing model responses to linguistic perturbations. Hashimoto et al. (2019) unified human and statistical evaluation for text generation.

**Diversity metrics: lexical vs semantic vs dynamical.** Prior diversity metrics can be categorized by what they measure:

- **Lexical**: distinct-$n$ (Li et al., 2016) and self-BLEU (Zhu et al., 2018) measure vocabulary variety across generated texts—counting unique $n$-grams or comparing pairwise similarity at the surface level.

- **Semantic**: cluster-based diversity (Hashimoto et al., 2019), semantic entropy (Kuhn et al., 2023), and embedding-space coverage metrics measure variety in *meaning* rather than surface form. These capture whether outputs span diverse semantic regions.

- **Dynamical (ours)**: Markov observables measure how diversity unfolds *over time*—not just "how varied are outputs?" but "how do conversations move between varied states?"

Our entropy rate relates to semantic diversity but captures temporal structure: a model that cycles through 5 states in fixed order has low entropy rate despite visiting diverse states. Spectral gap and stationary distribution have no direct analogues in static diversity metrics—they characterize mixing dynamics and equilibrium behavior that emerge only in multi-turn trajectories. Our contribution extends prior work by providing *dynamical* behavioral characterization: analyzing how conversations evolve over time rather than measuring static properties of individual responses.

## 2.3 Embedding Spaces and Clustering

Our framework relies on mapping text to continuous embedding space, then clustering to discrete states. Modern embedding models such as Sentence-BERT (Reimers & Gurevych, 2019), SimCSE (Gao et al., 2021), and instruction-tuned variants (Su et al., 2022) provide semantically meaningful representations where geometric proximity reflects semantic similarity.

Vector quantization—partitioning continuous spaces into discrete regions—has a long history in signal processing (Lloyd, 1982; Gray, 1984) and clustering (Arthur & Vassilvitskii, 2007). Classical approaches optimize reconstruction error, minimizing distortion between original vectors and their quantized representations. Our application imposes different requirements: we need *coverage* of the conversational space to ensure all semantic regions are represented, even when observations are sparse.

We employ a coverage-oriented variant based on farthest-point sampling (Gonzalez, 1985), which guarantees that every observation lies within a controlled distance of some cluster center. This approach trades optimal reconstruction for explicit coverage guarantees, ensuring that the discrete state space adequately represents the full range of conversational situations rather than concentrating on high-density regions.

# 3 Conversational Markov Chain Framework

This section presents our framework for analyzing LLM conversations as Markov chains. We begin with intuitive motivation for each technical component, then provide formal definitions.

## 3.1 From Conversations to Semantic States

**Embedding and clustering.** We map conversations to discrete semantic states through a two-step process. First, modern text embeddings map utterances to high-dimensional vectors where geometric proximity reflects semantic similarity. Two questions about atmospheric pressure, phrased differently, map to nearby vectors; a question about pressure and one about wildlife map far apart. This continuous representation captures semantic structure: while the embedding space has finite dimensions (e.g., 768), each dimension takes continuous real values, creating uncountably many possible representations.

We reduce to a finite state space through clustering. Given a collection of embedding vectors, we partition them into groups—each group defining a discrete "state" that represents a type of conversational situation. The number of states controls granularity: more states preserve finer distinctions but require more data for reliable analysis.

**Why semantic regions?**   Clustering in embedding space is a principled choice for state space construction. Embedding models trained on large corpora learn semantic representations that capture functional similarity—utterances serving similar conversational purposes map to nearby points (Mikolov et al., 2013; Devlin et al., 2019). Clustering these representations naturally groups functionally equivalent turns, creating states that reflect behavioral patterns rather than arbitrary text features. This approach has proven effective for discourse analysis (Li et al., 2016) and conversation modeling (Serban et al., 2016), where semantic coherence within states is crucial for meaningful analysis. Alternative state definitions (e.g., based on syntactic features or topic models) lack the robustness of learned embeddings across diverse conversational contexts.

**Formal definitions.**   Let $\mathcal{T}$ denote the space of text utterances. An *embedding function* $\phi : \mathcal{T} \to \mathbb{R}^d$ maps text to $d$-dimensional vectors. We assume embeddings are normalized to unit length, so that cosine similarity equals dot product.

A *quantization function* $q : \mathbb{R}^d \to \{1, \ldots, m\}$ assigns each embedding to one of $m$ discrete states. The complete mapping from text to states is the composition $\psi = q \circ \phi$:

$$\psi : \mathcal{T} \xrightarrow{\phi} \mathbb{R}^d \xrightarrow{q} \{1, \ldots, m\} \tag{1}$$

**Coverage-oriented quantization.**   Classical clustering (e.g., $k$-means) optimizes reconstruction error, concentrating cluster centers in high-density regions. For conversational analysis, we prefer *coverage*: ensuring every observed utterance lies within a controlled distance of some cluster center, even in sparsely populated semantic regions.

We employ farthest-point sampling (FPS) (Gonzalez, 1985): starting from an arbitrary point, iteratively add the point farthest from all existing centers until every observation is within radius $r$ of some center. This guarantees coverage while remaining computationally tractable. Once farthest-point sampling determines the number of clusters, we refine the centroids using Lloyd's algorithm (Lloyd, 1982) to reduce distortion within each cluster. The radius $r$ controls granularity: smaller $r$ yields more states (finer resolution) but requires more data. Algorithm details appear in Appendix B.

**Why FPS over k-means?**   Farthest-point sampling provides a deterministic bound on the maximum distance from any utterance to its nearest centroid, ensuring the state space spans the full embedding manifold rather than concentrating representational capacity in high-density regions. This coverage property is important for conversational analysis, where sparse but behaviourally significant semantic regions must be represented. While $k$-means clustering could also produce valid state spaces, it optimizes reconstruction error and may leave low-density regions unrepresented.

### 3.2   From State Sequences to Transition Matrices

Given a mapping from utterances to states, each conversation becomes a sequence of states: $(s_1, s_2, \ldots, s_n)$ where $s_i \in \{1, \ldots, m\}$. To characterize how conversations move through this state space, we construct an empirical transition matrix from observed state-to-state movements.

**Counting transitions.**   The simplest characterization of movement is a *transition matrix*: for each pair of states $(i, j)$, count how often conversations transition from state $i$ to state $j$. Normalizing each row to sum to one yields transition probabilities.

Formally, let $N_{ij}$ denote the number of observed transitions from state $i$ to state $j$ across all conversations. The empirical transition matrix is:

$$P_{ij} = \frac{N_{ij} + \alpha}{\sum_{j'=1}^{m} (N_{ij'} + \alpha)} \tag{2}$$

where $\alpha > 0$ is a smoothing parameter that prevents zero probabilities for unobserved transitions. This Dirichlet smoothing ensures the resulting Markov chain is ergodic—it can eventually reach any state from any starting point. We count transitions only *within* individual conversations, not across conversation boundaries—the transition from the last utterance of one conversation to the first of another carries no semantic meaning.

### 3.3 The Markov Property

**What makes it Markov?** A Markov chain has the property that the probability of the next state depends only on the current state, not on the full history:

$$P(s_{t+1} \mid s_t, s_{t-1}, \ldots, s_1) = P(s_{t+1} \mid s_t) \tag{3}$$

**Experimental design and the Markovian projection.** Our experimental design enforces first-order dependence at the generation level. Each LLM generation receives only the previous response as input—no accumulated conversation history is maintained. The model is called via Ollama's generate API without persistent context, ensuring that the only information available at turn $t$ is the content of turn $t-1$.

However, the quantization from continuous embeddings to discrete states is a non-invertible projection: the composition $\psi = q \circ \phi$ (text $\rightarrow$ embedding $\rightarrow$ discrete state) is many-to-one, so the projected state sequence is not guaranteed to be exactly first-order Markov—this is the classical lumpability problem (Kemeny & Snell, 1960). Section C.5 quantifies the fidelity of the first-order approximation (82–83% of transition structure captured). We therefore treat the method as analyzing a *Markovian projection* of the conversation: the transition matrix captures first-order sequential structure, and the comparative statistics derived from this projection remain well-defined and yield valid between-model comparisons.

**When the property holds.** The Markov property holds precisely when:

- **Context is limited to the previous turn.** Our generation protocol passes only the immediately preceding response, not conversation history.

- **States capture relevant semantic content.** The clustering assigns semantically similar responses to the same state, so knowing the state captures the information available to the next generation.

- **State granularity is appropriate.** The choice of $m$ states balances two competing objectives: finer granularity (more states) reduces distortion by preserving semantic distinctions, but introduces data sparsity that undermines statistical confidence in transition estimates. Coarser granularity (fewer states) aggregates more variation within clusters but ensures sufficient observations per state for reliable estimation. At $m = 25$ states, sufficient observations per state ensure reliable transition probability estimates while capturing meaningful behavioral distinctions—the first-order model explains 82–83% of observed transition structure (Appendix C.5).

**Scope.** Our experimental regime demonstrates that the Markov framework yields meaningful, distinguishing behavioral signatures under controlled conditions. The results validate the framework's utility for behavioral characterization. In experimental regimes with richer context (e.g., full conversation history passed to the model), higher-order dependencies may become more significant relative to the quantified observables.

### 3.4 Time Homogeneity

**The assumption.** A time-homogeneous Markov chain assumes that transition probabilities remain constant over time: $P(s_{t+1} = j \mid s_t = i)$ does not depend on $t$ (Levin et al., 2017). This is distinct from the Markov property itself (which concerns conditional independence from history) and requires that the stochastic process governing transitions remains stationary throughout data collection.

**Why approximate?** We term this property *approximate* for two reasons. First, our experimental design uses batch generation across multiple sessions, introducing potential variation from model loading, quantization differences, or hardware states. Second, even under ideal conditions, any finite dataset yields empirical transition matrices that deviate from true underlying probabilities due to sampling variation. The question is whether systematic drift (genuine non-stationarity) is distinguishable from sampling noise.

**Empirical validation.** We assess time homogeneity by partitioning conversations into temporal segments and comparing transition matrices across segments using total variation distance (Levin et al., 2017). For our datasets, segment-to-segment variation remains below 0.15, well within the range expected from sampling variation with $N \approx 100,000$ transitions per segment (Appendix C.9). Systematic drift would manifest as monotonic trends in specific transition probabilities over time; we observe no such patterns.

**Impact of violations.** Violations of time homogeneity would inflate variance in our observables and potentially introduce bias if drift patterns differ between models. For coarse behavioral characterization ("Model A is more exploratory than Model B"), our observed deviations (total variation distance $< 0.15$) have negligible impact on rank-order comparisons. For fine-grained analysis of specific transition probabilities, stricter homogeneity would be required. Our empirical validation targets the former use case, where approximate stationarity suffices for meaningful comparison.

### 3.5 Sample Complexity Considerations

**The fundamental trade-off.** Finer state-space resolution preserves more behavioral detail but requires more data for reliable estimation. Each row of the transition matrix contains $m$ probabilities that must be estimated from observations of that row's source state. Standard concentration inequalities (Levin et al., 2017) indicate that reliable estimation of $m$ probabilities requires $\Omega(m \ln m)$ samples, where ln denotes natural logarithm.

**Quadratic scaling.** Since we have $m$ rows, and observations distribute across rows according to state occupancy, the total sample requirement scales as $N = \Omega(m^2 \ln m)$. This is a fundamental barrier: doubling the number of states quadruples the data requirement.

**Sparsity considerations.** For $m = 25$ states (our case study), the naive bound assuming uniform state visitation is $N \approx 2,000$. However, when stationary distributions are concentrated—as they are in conversational data, where some semantic states are visited far more frequently than others—rare states require disproportionately more total samples. Accounting for observed sparsity patterns in our data ($\pi_{\min} \approx 0.02$ for the rarest states), stable estimates require $N \gtrsim 100,000$ transitions. Our datasets ($N \approx 117,500$ for Llama, $N \approx 200,000$ for Mistral) meet this requirement with modest safety margins (Appendix C.8 provides detailed analysis). For $m = 100$ states under similar sparsity, the requirement would grow to $N \gtrsim 1,000,000$ or more. This sample complexity positions our framework as a *targeted tool* for applications willing to invest in substantial data collection, not a lightweight metric applicable to small datasets.

**Practical implications.** In our experiments, we generate over 300,000 conversation turns to achieve stable estimates with $m = 25$ states. This represents approximately 50 hours of LLM generation time and substantial computational cost. The investment is justified for critical applications where behavioral characterization matters, but precludes casual application to small-scale experiments.

## 4 Behavioral Observables

Given a transition matrix $P$, classical Markov chain theory provides several quantities that characterize the chain's dynamics. We focus on three observables that admit natural behavioral interpretations: stationary distribution, entropy rate, and spectral gap. Each captures a different aspect of conversational dynamics.

Throughout this section, we use the term *behavioral modes* to refer to distinct response patterns or conversational styles exhibited by the model—different types of behavior characterized by which semantic regions (states) the model occupies and how it transitions between them. A model may exhibit exploratory modes (diverse transitions across many states), stereotyped modes (repetitive patterns within few states), or persistent modes (prolonged dwelling in specific regions). Our observables quantify these behavioral distinctions.

### 4.1 Stationary Distribution: Where Does the Conversation Tend to Go?

Over many conversations, or equivalently over many turns of a single long conversation, some semantic states are visited frequently while others are rare. The stationary distribution captures this long-run occupancy pattern.

A uniform stationary distribution means the model visits all semantic regions equally—it has a broad repertoire and doesn't favor particular types of exchanges. A concentrated distribution means the model spends most of its time in a few dominant states—it has a narrow repertoire with "attractor" states that conversations repeatedly visit.

**Formal definition.** The stationary distribution $\pi \in \mathbb{R}^m$ satisfies:

$$\pi^\top P = \pi^\top, \quad \sum_{i=1}^m \pi_i = 1, \quad \pi_i \geq 0 \tag{4}$$

For an ergodic chain (guaranteed by our smoothing), $\pi$ is unique and equals the limiting distribution: $\lim_{t\to\infty} P^t(i, \cdot) = \pi$ for any initial state $i$.

**Characterizing concentration.** We summarize the stationary distribution through two quantities:

- **Peak concentration:** $\|\pi\|_\infty = \max_i \pi_i$, the probability mass of the most-visited state. Higher values indicate stronger concentration.

- **Stationary entropy:** $H(\pi) = -\sum_i \pi_i \log_2 \pi_i$, measuring how evenly distributed the mass is. Higher values indicate broader repertoire.

**Behavioral interpretation.** A model with high peak concentration and low stationary entropy exhibits narrow behavioral repertoire: conversations repeatedly return to a small set of dominant states. This may indicate "mode collapse" where the model favors certain response patterns regardless of context. A model with low peak concentration and high stationary entropy exhibits broad repertoire: the model engages with the full range of semantic territory available in its conversational regime.

### 4.2 Entropy Rate: How Unpredictable Is the Next Move?

Suppose you observe a conversation in progress and try to predict what type of situation comes next. Sometimes the prediction is easy—if the model tends to follow questions with explanations, you can anticipate explanations after questions. Sometimes prediction is hard—if the model transitions to many different states with similar probability, any guess is as good as another.

Entropy rate quantifies this predictability. High entropy means transitions are unpredictable; the conversation could go many directions. Low entropy means transitions are predictable; the conversation follows well-worn paths.

**Formal definition.** The entropy rate of a Markov chain measures the average uncertainty in the next state, where the average is taken over the long-run distribution of current states:

$$h = -\sum_{i=1}^m \sum_{j=1}^m \pi_i P_{ij} \log_2 P_{ij} \tag{5}$$

Here $\pi$ is the stationary distribution defined above, and the convention is $0 \log 0 = 0$. The entropy rate is measured in bits.

**Behavioral interpretation.** A model with high entropy rate exhibits diverse, exploratory conversational behavior. Given the current semantic state, many next states have non-negligible probability; the model

doesn't converge to stereotyped patterns. A model with low entropy rate behaves predictably—once you know the current state, you can anticipate the next move with high confidence.

In practical terms: if $h \approx 3$ bits, the effective "branching factor" is roughly $2^3 = 8$—the model might plausibly transition to about 8 different states from any given state. If $h \approx 2$ bits, the effective branching factor is about 4. This provides an interpretable scale for diversity.

### 4.3 Spectral Gap: How Persistent Are Conversational Patterns?

Start a conversation from different initial states—perhaps one begins with a factual question, another with a hypothetical. Over time, do these conversations converge to similar patterns, or do they remain distinct?

The spectral gap measures how quickly conversations reach their equilibrium behavior. A large spectral gap means rapid equilibration: after a few turns, the distribution over states is nearly independent of where the conversation began. A small spectral gap means slow equilibration: the conversation dwells in its starting region for many turns before mixing with other trajectories.

**Formal definition.** Let $\lambda_1 = 1 > |\lambda_2| \geq \cdots \geq |\lambda_m|$ be the eigenvalues of the transition matrix $P$, ordered by magnitude. The spectral gap is:

$$\gamma = 1 - |\lambda_2| \tag{6}$$

The spectral gap controls the mixing time—the number of steps required for the state distribution to approach the stationary distribution regardless of initial state. For reversible Markov chains, tight bounds relate mixing time to spectral gap (Levin et al., 2017):

$$\tau_{\text{mix}}(\epsilon) \leq \frac{\ln(1/(\epsilon \pi_{\min}))}{\gamma} \tag{7}$$

where $\pi_{\min} = \min_i \pi_i$ is the minimum stationary probability and ln denotes natural logarithm. For general ergodic chains (which may not satisfy reversibility), the relationship is more complex, but the characteristic time scale $1/\gamma$ remains a valid measure of convergence speed: larger spectral gap indicates faster equilibration. We use $1/\gamma$ for practical interpretation, which provides the order-of-magnitude mixing rate for comparative analysis.

**Behavioral interpretation.** The spectral gap reflects the balance between persistence and responsiveness in the model's conversational flow. A model with large spectral gap exhibits "responsive" dynamics: it moves freely between behavioral modes and adapts readily to different semantic regions. A model with small spectral gap exhibits "persistent" dynamics: once engaged in a particular mode, it remains there for many turns before shifting. High-gap dynamics move freely between behavioral modes; low-gap dynamics remain in the same mode for longer stretches.

For example, using the characteristic time scale: if $\gamma = 0.25$, then $1/\gamma = 4$ steps characterizes the mixing rate. If $\gamma = 0.10$, then $1/\gamma = 10$ steps—persistence is more than twice as strong.

### 4.4 Relationship to Correctness Metrics

These behavioral observables provide a *complementary* perspective to accuracy-focused evaluation, not a replacement. Two models with identical benchmark accuracy may differ substantially in behavioral dynamics:

- **Same accuracy, different entropy:** One model gives correct answers through diverse reasoning paths; another gives correct answers through memorized templates. Both score equally on accuracy, but differ in entropy rate.

- **Same accuracy, different mixing:** One model readily shifts between topics when prompted; another persists in the current topic even when asked to change. Both may answer questions correctly, but differ in spectral gap.

- **Same accuracy, different repertoire:** One model engages broadly across the conversational domain; another concentrates on a subset of familiar territory while still answering correctly when prompted. Both score equally, but differ in stationary distribution.

The appropriate choice depends on application requirements. A standardized documentation system may prefer low entropy and strong concentration (predictable, template-adherent behavior). An exploratory tutoring system may prefer high entropy and broad repertoire (diverse, adaptive behavior). Our framework provides the quantitative tools to distinguish these behavioral modes.

## 5 Methods

### 5.1 Domain and Models

We study teacher-student conversations about climate and desert formation. The domain involves physical mechanisms (pressure systems, evaporation, trade winds) with diverse exchange types: factual questions, clarification requests, analogies, and explanations. The domain allows for natural conversational variety, and the prompting structure permits revisiting of earlier topics within the same conversation, contributing to the irreducible character of the resulting Markov chains.

We compare two language models:

- **Llama 3.1 8B** (Touvron et al., 2023): Meta's open-weight instruction-tuned model (8 billion parameters, checkpoint: `llama3.1:latest`, Q4_K_M quantization, downloaded November 2025)
- **Mistral 7B v0.1** (Jiang et al., 2023): Mistral AI's instruction-tuned model (7 billion parameters, checkpoint: `mistral:latest`, Q4_K_M quantization, downloaded November 2025)

Both models are served locally via Ollama v0.13.1 (`https://ollama.com`), a local LLM serving framework. Complete model specifications including SHA256 digests appear in Appendix D.1.

### 5.2 Data Generation

We generate conversations using two instances of the same LLM, differing only in their system prompts: one acts as a "teacher" and one as a "student." The output from one instance is fed directly into the other as the sole context—no conversation history or additional prompting is retained between turns. This stateless design ensures each utterance depends only on the immediately preceding turn, supporting the Markov assumption. The results reported here characterise model behaviour under this minimal role-only prompting regime; sensitivity of the observables to alternative prompt configurations is a natural follow-up application of the framework.

Each conversation comprises exactly 5 turn pairs (10 total utterances) focused on climate and desert formation. The verbatim system prompts and complete protocol details appear in Section 5.2.1 and Appendix D.3.

Dataset sizes:

- Llama 3.1 8B: 117,524 utterances across thousands of conversations
- Mistral 7B: 200,000 utterances across thousands of conversations

#### 5.2.1 Generation Protocol

Each conversation is initiated by the teacher instance with a fixed seed question about desert formation (e.g., "Why are large deserts commonly found around 30 degrees north and south of the equator?"). The student instance responds, and the exchange continues for exactly 5 turn pairs. The teacher's system prompt instructs it to ask context-rich follow-up questions; the student's prompt elicits typical high-school student responses with appropriate uncertainty. Both instances use identical sampling parameters to ensure comparability.

Sampling parameters (identical for both models):

- Temperature: 0.40

- Top-$p$: 0.70

- Top-$k$: 50

- Global random seed: 42

- Repeat penalty: 1.15 (last 32 tokens)

Complete sampling parameters and verbatim system prompts appear in Appendix D. Conversations terminate after exactly 5 exchange pairs.

Experiments were conducted November 3–6, 2025 on local hardware using Ollama's stateless generation API (`num_keep=0`), ensuring each generation received only the immediately preceding utterance.

### 5.2.2   Embedding

Each utterance is embedded using `nomic-embed-text:latest` (Nussbaum et al., 2024) via Ollama v0.13.1 (downloaded November 2025), producing 768-dimensional vectors normalized to unit length. The embedding model SHA256 digest and complete specifications appear in Appendix D.1.

### 5.3   Quantization Pipeline

Farthest-point initialization: starting from a deterministic seed point, we iteratively add the farthest point from all existing centers until coverage is achieved within the target radius. Lloyd refinement (Lloyd, 1982) is optionally applied to reduce average distortion, with reversion if refinement degrades coverage or creates empty clusters.

We construct the quantization independently for each model: FPS clustering is applied to Llama's conversation embeddings to determine its state space ($m = 25$ states), and independently to Mistral's embeddings to determine its separate state space ($m = 25$ states). This per-model design ensures that behavioural observables reflect intrinsic model properties rather than artefacts of shared discretisation. Dirichlet smoothing parameter $\alpha = 0.5$ (Jeffreys prior).

### 5.4   Statistical Methods

Uncertainty in Markov observables is estimated through bootstrap resampling: conversations are resampled with replacement and observables recomputed (1000 iterations). The standard deviation of the distribution provides uncertainty estimates.

Significance is assessed via bootstrap 95% confidence intervals (Efron, 1979) and permutation testing (Good, 2005). Bootstrap methods are non-parametric and suitable for complex Markov chain statistics (entropy rate, spectral gap, stationary distribution) where analytical formulas for sampling variance are intractable. The bootstrap captures uncertainty arising from the finite sample of conversations by resampling observed data, making no distributional assumptions about the observables.

Permutation testing provides distribution-free hypothesis testing by comparing observed differences to a null distribution generated by randomly permuting model labels. This approach tests whether the two models' observables come from the same distribution without requiring parametric assumptions about the null sampling distribution.

Effect sizes are quantified using Cohen's $d$ (Cohen, 1988) (standardized mean difference) and relative percentage differences. Cohen's $d$ provides a scale-independent measure of effect magnitude, enabling comparison across observables with different units (bits for entropy, dimensionless for spectral gap). Following conventional interpretations, we classify effects as negligible ($|d| < 0.2$), small ($0.2 \leq |d| < 0.5$), medium ($0.5 \leq |d| < 0.8$), or large ($|d| \geq 0.8$). Unlike $p$-values which depend on sample size, effect sizes characterize practical importance: our observed values ($d > 5$) indicate substantial behavioral distinctions.

## 6 Results

### 6.1 Model Comparison

Table 1 summarizes the behavioral observables for both models under identical experimental conditions ($m = 25$ states, $\alpha = 0.5$ smoothing).

Table 1: Behavioral observables for Llama 3.1 8B and Mistral 7B on teacher-student dialogues. Uncertainties are bootstrap standard errors.

| Observable | Llama 3.1 | Mistral 7B |
|---|---|---|
| Entropy rate $h$ (bits) | $2.91 \pm 0.03$ | $2.46 \pm 0.02$ |
| Spectral gap $\gamma$ | $0.28 \pm 0.02$ | $0.098 \pm 0.01$ |
| Peak concentration $\|\pi\|_\infty$ | 10.5% | 17.3% |
| Stationary entropy $H(\pi)$ (bits) | 4.21 | 3.83 |

In our teacher-student dialogue experiments, Llama exhibits 18% higher entropy rate ($h = 2.91$ vs. $h = 2.46$ bits), corresponding to effective branching factors of $2^{2.91} \approx 7.5$ and $2^{2.46} \approx 5.5$ respectively.

Llama's spectral gap is nearly three times larger than Mistral's ($\gamma = 0.28$ vs. $\gamma = 0.098$), corresponding to characteristic mixing time scales of $1/\gamma \approx 3.6$ and $1/\gamma \approx 10.2$ turns respectively.

Mistral concentrates 65% more probability mass in its most-visited state (17.3% vs. 10.5%), with 9% lower stationary entropy (3.83 vs. 4.21 bits).

These orderings are robust across state counts $m \in \{15, 25, 50\}$ (Appendix C.6). These results characterise how the models react under identical experimental conditions (fixed sampling parameters, minimal prompting, stateless generation). Sensitivity of the observables to parameter variation is a natural direction for follow-up work.

### 6.2 Statistical Significance

Bootstrap 95% confidence intervals do not overlap for any primary observable:

- Entropy rate: Llama $[2.85, 2.97]$ vs. Mistral $[2.42, 2.50]$

- Spectral gap: Llama $[0.26, 0.30]$ vs. Mistral $[0.09, 0.11]$

- Peak concentration: Llama $[9.8\%, 11.2\%]$ vs. Mistral $[16.5\%, 18.1\%]$

All differences achieve $p < 0.001$ based on permutation testing.

Effect sizes:

- Entropy rate: Cohen's $d > 5$

- Spectral gap: factor of $\sim 3\times$

- Peak concentration: 65% relative increase

The entropy rate difference ($\Delta h = 0.45$ bits) is approximately 15 times the combined measurement uncertainty ($\sigma \approx 0.03$ bits).

### 6.3 Visualizations

Figure 1 shows transition matrix heatmaps for both models. These visualizations provide direct insight into the behavioral differences quantified by our three observables.

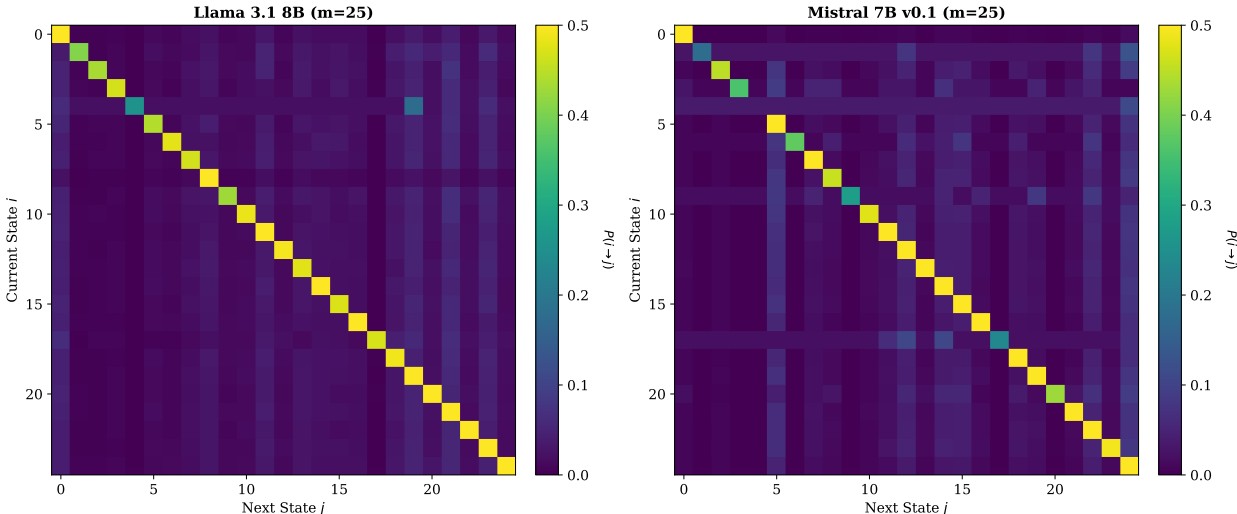

Figure 1: Transition matrices for Llama 3.1 (left) and Mistral 7B (right). Each cell $(i, j)$ shows the probability of transitioning from state $i$ to state $j$. Llama exhibits more uniform transition probabilities across many states (lighter overall shading), while Mistral concentrates probability mass in specific transitions (darker cells indicating higher probabilities). These structural differences manifest as the quantitative distinctions in entropy rate, spectral gap, and stationary distribution reported in Table 1. Mechanistic interpretation of these behavioral differences is left for future work.

**Reading the transition matrices.** Each matrix is a $25 \times 25$ heatmap where rows and columns represent the semantic states discovered through clustering. Cell shading intensity (darker = higher probability) shows how likely the model is to transition from one state to another. The diagonal represents self-transitions (remaining in the same semantic region for consecutive turns), while off-diagonal elements show movements between different regions.

**Key patterns to notice.** Llama's matrix exhibits relatively uniform shading across many cells, indicating that from any given state, the model has non-negligible probability of transitioning to many different states. This visual diffuseness directly corresponds to the high entropy rate ($h = 2.91$ bits). In contrast, Mistral's matrix shows concentrated darker patches—specific state pairs with dominant transition probabilities—surrounded by lighter regions. This concentration reflects the lower entropy rate ($h = 2.46$ bits) and narrower stationary distribution.

The spectral gap difference is also visible: Llama's more uniform matrix implies weaker persistence (transitions spread across the state space), while Mistral's concentrated structure implies stronger persistence (transitions clustered around specific pathways). These visual patterns provide intuitive support for the quantitative measurements in Table 1.

# 7 Discussion

## 7.1 Interpretation of Results

**Behavioral profiles.** Figure 2 summarizes the three behavioral observables for both models. The observables paint coherent behavioral profiles for each model:

- **Llama:** Exploratory, diverse, fluid. High entropy rate indicates unpredictable transitions; large spectral gap indicates rapid topic shifts; broad stationary distribution indicates even coverage of semantic territory.

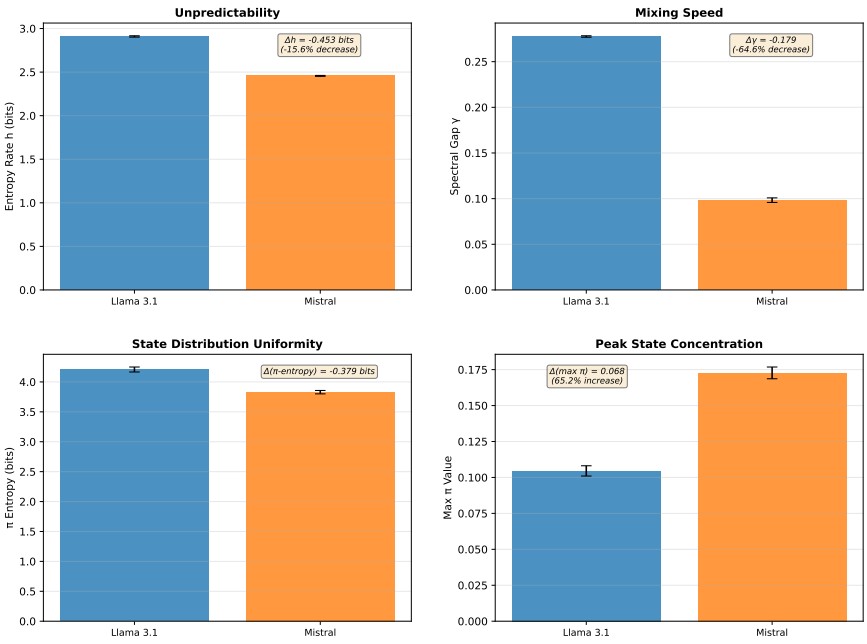

Figure 2: Behavioral observables comparison showing the three complementary metrics that characterize conversational dynamics. Left: Entropy rate (bits) measures transition diversity—Llama's higher value indicates more unpredictable, varied conversational flow. Center: Spectral gap (dimensionless) measures responsiveness—Llama's threefold larger gap indicates faster equilibration and more fluid topic shifts. Right: Peak stationary concentration (%) measures repertoire breadth—Mistral's higher value indicates stronger preference for specific conversational patterns. Error bars show bootstrap 95% confidence intervals. All differences are statistically significant ($p < 0.001$) with large effect sizes (Cohen's $d > 5$ for entropy rate). The consistent ordering (Llama higher entropy/gap, lower concentration) reveals coherent behavioral profiles: Llama exhibits exploratory dynamics while Mistral exhibits stereotyped persistence.

- **Mistral:** Stereotyped, persistent, concentrated. Low entropy rate indicates predictable transitions; small spectral gap indicates prolonged dwelling; narrow stationary distribution indicates repeated revisiting of preferred states.

Neither profile is universally superior—appropriateness depends on application requirements.

**Entropy rate interpretation.** The 18% higher entropy rate for Llama ($h = 2.91$ vs. $h = 2.46$ bits) manifests as response diversity: from any given conversational state, Llama plausibly transitions to more distinct next states than Mistral. In our experimental domain, when given similar conversational contexts, Llama generates more varied formulations across trials, while Mistral converges to more stereotyped patterns. Llama explores semantic territory; Mistral follows well-worn paths.

**Spectral gap interpretation.** The threefold difference in spectral gap ($\gamma = 0.28$ vs. $\gamma = 0.098$) translates directly to mixing time differences. Llama transitions fluidly between semantic regions—moving readily from explaining pressure systems to discussing evaporation to addressing trade winds. Mistral persists: once engaged in a topic, it dwells there for many turns before transitioning elsewhere. This persistence may be advantageous for in-depth exploration of a single topic but disadvantageous for conversations requiring agile topic shifts.

**Stationary distribution interpretation.** Mistral's 65% higher peak concentration (17.3% vs. 10.5%) indicates that Mistral's conversations repeatedly return to a small set of "attractor" states—particular types

of explanations or response patterns that the model favors regardless of context. Llama distributes attention more evenly across the semantic landscape, exhibiting a broader behavioral repertoire.

## 7.2 Methodological Considerations

**Methodological choices.** The framework's measurements depend on embedding model (`nomic-embed-text`) and clustering parameters (radius, initialization, state count). Different choices yield different state spaces and potentially different observables—results describe behavior *as viewed through this analytical lens*. When applying the framework to new domains, sensitivity analysis across state counts helps assess robustness of conclusions.

**Spectral gap interpretation.** Classical spectral gap theory provides tight mixing time bounds for reversible Markov chains. Conversational chains are likely non-reversible due to inherent directionality in dialogue (questions prompt explanations differently than explanations prompt questions). For non-reversible chains, the spectral gap remains a valid comparative measure of mixing speed—models with larger spectral gap equilibrate faster—though the precise relationship to mixing time is more complex. We use spectral gap primarily for relative comparison between models rather than absolute mixing time prediction.

**Framework applicability and scope.** The Markov chain framework applies when experimental conditions create approximate Markov behavior: limited context windows, appropriate state granularity, and relatively stationary dynamics. LLMs do not universally behave as Markov chains—the methodology analyzes a Markovian projection of their conversational behavior, and controlled experimental conditions (such as stateless generation) improve the fidelity of this projection.

Our measurements characterize behavior *within the specific conversational domain and semantic space we analyzed*—teacher-student dialogues about atmospheric science. We do not claim to characterize "Llama 3.1" or "Mistral 7B" universally across all possible uses. This specificity is a feature: the framework is one tool in a wider set of measurement techniques for describing LLM behaviour—complementing, not replacing, existing evaluation methods. Practitioners can apply it to characterize LLM behavior in their target domain or use case, obtaining context-relevant measurements rather than broad generalizations that may not apply to their application.

Different conversational contexts (topics, styles, domains, conversation structures) may yield different behavioral signatures. This variability is expected and reflects LLMs' context-dependent behavior—it does not invalidate the framework. Rather, it demonstrates the framework's ability to detect domain-specific behavioral patterns. The finding that "Llama is more exploratory than Mistral in atmospheric science tutoring" is a precise, actionable statement for practitioners in that domain. Whether this ordering holds in creative writing or code generation requires empirical investigation in those contexts. This context-specificity reinforces the value of domain-targeted behavioral analysis over coarse universal claims.

## 7.3 Potential Applications

The framework enables several types of analysis unavailable from accuracy-focused evaluation:

**Model selection for behavioral requirements.** When applications require specific behavioral properties—high diversity for creative tasks, high predictability for standardized documentation—the observables provide quantitative guidance.

**Prompt engineering validation.** The framework can validate whether prompting interventions achieve intended effects: does adding "be creative" actually increase entropy rate? Does providing templates decrease it?

**Fine-tuning drift monitoring.** Tracking Markov observables over model versions provides early warning of behavioral changes that might not surface in accuracy benchmarks.

**Safety and robustness analysis.** Behavioral concentration could indicate mode collapse or vulnerability to adversarial steering. Models that persist in semantic regions may be more susceptible to manipulation exploiting this persistence.

### 7.4 When to Use This Framework

**Appropriate contexts:**

- Critical applications where behavioral characterization matters beyond accuracy

- Sufficient resources for large-scale data collection

- Interest in dynamics and trajectories, not just individual responses

- Comparison of models or configurations on matched conversational regimes

**Inappropriate contexts:**

- Lightweight, rapid model evaluation (use benchmarks instead)

- Small-scale experiments with limited data

- Single-turn tasks where conversational dynamics are irrelevant

- Situations requiring absolute behavioral claims rather than relative comparisons

The framework is a specialized instrument for deep behavioral analysis, not a general-purpose evaluation metric.

## 8 Conclusion

We have presented a framework for characterizing LLM conversational behavior through Markov chain analysis over semantic states. By mapping conversations to discrete states via embedding and clustering, we extract three interpretable observables—entropy rate, spectral gap, and stationary distribution—that capture distinct aspects of behavioral dynamics invisible to accuracy-focused evaluation.

Our case study comparing Llama 3.1 8B and Mistral 7B in teacher-student dialogues demonstrates that the framework detects meaningful behavioral differences: Llama exhibits higher entropy (diverse responses), faster mixing (fluid topic transitions), and broader stationary distribution (even semantic coverage), while Mistral shows lower entropy (predictable patterns), slower mixing (persistent dwelling), and concentrated stationary distribution (attractor states).

The framework fills a gap in LLM evaluation methodology. Accuracy benchmarks tell us what models know; our framework tells us how they behave. For applications where behavioral dynamics matter—diversity, persistence, exploration patterns—this provides principled, quantitative tools that complement existing evaluation paradigms.

**Future directions.** Several extensions merit investigation: (1) adaptive quantization that adjusts state granularity to data density, (2) multi-scale analysis combining coarse and fine state representations, (3) cross-domain validation to characterize generalization of behavioral findings, (4) streaming algorithms for efficient computation as conversations accumulate, and (5) investigation of richer-context experimental regimes where higher-order dependencies may become more significant.

**Reproducibility.** To ensure full reproducibility of our results, all implementation code and analysis scripts will be publicly available at `https://github.com/xxx/xxx` upon publication.

The contribution of this work is the framework itself—a proof-of-concept that conversational dynamics can be characterized through Markov chain analysis. We hope this opens avenues for behavioral evaluation that go beyond measuring what models get right, toward understanding how they engage in the dynamic process of conversation.

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

## A Proofs and Derivations

This appendix provides formal proofs and derivations for the theoretical claims in the main text.

### A.1 Coverage Guarantee

**Proposition 1** (Coverage Guarantee). *For any set of unit-normalized embeddings $\mathbf{X} = \{\mathbf{x}_1, \ldots, \mathbf{x}_N\}$ and radius $r > 0$, Algorithm 1 (farthest-point traversal) produces a centroid set $\mathbf{C}$ such that:*

$$\max_{\mathbf{x} \in \mathbf{X}} \min_{\mathbf{c} \in \mathbf{C}} (1 - \mathbf{x}^\top \mathbf{c}) \leq r \tag{8}$$

*Proof.* The algorithm terminates when $\max_i d_i \leq r$, where $d_i = \min_{\mathbf{c} \in \mathbf{C}} (1 - \mathbf{x}_i^\top \mathbf{c})$ is the distance from $\mathbf{x}_i$ to its nearest centroid. By construction, the algorithm only terminates when this maximum distance condition is satisfied. Since we add points as centroids until coverage is achieved, and each addition strictly reduces the maximum uncovered distance (by selecting the farthest point), the algorithm must terminate with the coverage guarantee satisfied. □

### A.2 Sample Complexity Bound

**Proposition 2** (Sample Complexity). *To estimate each row of the transition matrix $P$ within accuracy $\epsilon$ with probability $1 - \delta$, we require:*

$$N_i \geq \frac{m \ln(m/\delta)}{2\epsilon^2} \tag{9}$$

*observations from state $i$, where $\ln$ denotes natural logarithm. Under uniform state visitation, this yields a global requirement of $N = \Omega(m^2 \ln m)$.*

*Proof.* Consider row $i$ of the transition matrix, consisting of $m$ probabilities $\{P_{ij}\}_{j=1}^m$ that sum to 1. From $N_i$ independent transitions starting from state $i$, we estimate each $P_{ij}$ by the empirical frequency $\hat{P}_{ij} = N_{ij}/N_i$, where $N_{ij}$ is the count of transitions to state $j$.

By Hoeffding's inequality, for any single transition probability:

$$\Pr(|\hat{P}_{ij} - P_{ij}| > \epsilon) \leq 2\exp(-2N_i\epsilon^2) \tag{10}$$

Applying a union bound over all $m$ target states:

$$\Pr(\|\hat{P}_i - P_i\|_\infty > \epsilon) \leq 2m\exp(-2N_i\epsilon^2) \tag{11}$$

Setting this failure probability to $\delta$ and solving for $N_i$:

$$N_i \geq \frac{\ln(2m/\delta)}{2\epsilon^2} = \Omega\left(\frac{m\ln(m/\delta)}{\epsilon^2}\right) \tag{12}$$

If states are visited with approximately uniform frequency (each visited $\Theta(N/m)$ times), then $N_i \approx N/m$, yielding the global bound $N = \Omega(m^2 \ln m)$. However, when the stationary distribution is concentrated and the minimum state probability $\pi_{\min}$ is small, the requirement scales as $N = \Omega(m^2 \ln m/\pi_{\min})$ (see Appendix C.8 for detailed analysis). □

## A.3 Ergodicity Under Dirichlet Smoothing

**Theorem 1** (Ergodicity). *For any $\alpha > 0$, the transition matrix $P$ defined by Dirichlet smoothing (Equation 4 in main text) corresponds to an ergodic Markov chain with a unique stationary distribution.*

*Proof.* The smoothed transition matrix has entries:

$$P_{ij} = \frac{N_{ij} + \alpha}{\sum_{j'}(N_{ij'} + \alpha)} \tag{13}$$

For $\alpha > 0$, every entry $P_{ij} > 0$ since the numerator includes $\alpha > 0$. Thus $P$ is a strictly positive stochastic matrix.

A strictly positive stochastic matrix is irreducible (every state is reachable from every other state in one step) and aperiodic (the GCD of return times to any state is 1, since $P_{ii} > 0$ allows returns in one step).

By the Perron-Frobenius theorem, an irreducible aperiodic stochastic matrix has a unique stationary distribution $\pi$ satisfying $\pi^\top P = \pi^\top$, and $\lim_{t\to\infty} P^t$ converges to the rank-1 matrix with all rows equal to $\pi$. □

## A.4 Spectral Gap and Mixing Time

**Proposition 3** (Mixing Time Bound for Reversible Chains). *For an ergodic reversible Markov chain with spectral gap $\gamma$, the mixing time satisfies:*

$$\tau_{mix}(\epsilon) \leq \frac{\ln(1/(\epsilon\pi_{\min}))}{\gamma} \tag{14}$$

*where $\pi_{\min} = \min_i \pi_i$ is the minimum stationary probability and $\ln$ denotes natural logarithm.*

*Proof.* See Levin, Peres, and Wilmer (Levin et al., 2017), Theorem 12.3. For reversible chains, the transition matrix can be symmetrized, yielding real eigenvalues $1 = \lambda_1 > \lambda_2 \geq \cdots \geq \lambda_m \geq -1$. The spectral gap $\gamma = 1 - \lambda_2$ controls the rate of convergence to stationarity. The total variation distance from stationarity decays as $\|P^t(i, \cdot) - \pi\|_{TV} \leq \sqrt{\frac{1}{\pi_i}}(1 - \gamma)^t$. Setting this bound to $\epsilon$ and solving for $t$ yields the mixing time bound. □

**General (non-reversible) chains.** For general ergodic Markov chains that may not satisfy reversibility (detailed balance: $\pi_i P_{ij} = \pi_j P_{ji}$), the spectral theory is more complex. Eigenvalues may be complex, and the second eigenvalue is defined as $\lambda_2 = \max\{|\lambda| : \lambda \text{ eigenvalue of } P, \lambda \neq 1\}$. The spectral gap $\gamma = 1 - |\lambda_2|$ still characterizes mixing speed, with $1/\gamma$ providing the characteristic time scale for convergence, though the logarithmic bound above may not hold exactly.

Conversational Markov chains derived from LLM interactions are likely non-reversible due to inherent directionality—the joint probability of observing a (question, explanation) pair differs from that of an (explanation, question) pair, i.e., the flow $\pi_i P_{ij}$ is not symmetric, so detailed balance does not hold. Nevertheless, the spectral gap remains a valid comparative measure: models with larger $\gamma$ equilibrate faster than those with smaller $\gamma$. Our primary use of spectral gap is for relative comparison between models rather than absolute mixing time prediction.

# B    Algorithms and Implementation

This appendix provides complete algorithmic details for the quantization and Markov construction pipeline.

## B.1    Farthest-Point Quantization

Algorithm 1 presents the complete farthest-point traversal procedure for coverage-oriented quantization.

---

**Algorithm 1** Farthest-Point Quantization

---

**Require:** Unit-normalized embeddings $\mathbf{X} = \{\mathbf{x}_1, \ldots, \mathbf{x}_N\} \subset \mathbb{R}^d$
**Require:** Radius parameter $r \in (0, 2]$
**Ensure:** Centroid set $\mathbf{C}$, assignment function $q$
 1: **Initialize:** Select seed $\mathbf{c}_1 = \mathbf{x}_{\arg\max_i \|\mathbf{x}_i\|_1}$ (deterministic tie-breaking)
 2: $\mathbf{C} \leftarrow \{\mathbf{c}_1\}$
 3: $d_i \leftarrow 1 - \mathbf{x}_i^\top \mathbf{c}_1$ for all $i \in \{1, \ldots, N\}$
 4: **while** $\max_i d_i > r$ **do**
 5:     $j^* \leftarrow \arg\max_i d_i$ (with deterministic tie-breaking by index)
 6:     $\mathbf{c}_{\text{new}} \leftarrow \mathbf{x}_{j^*}$
 7:     $\mathbf{C} \leftarrow \mathbf{C} \cup \{\mathbf{c}_{\text{new}}\}$
 8:     **for** $i = 1$ to $N$ **do**
 9:         $d_i \leftarrow \min(d_i, 1 - \mathbf{x}_i^\top \mathbf{c}_{\text{new}})$
10:     **end for**
11: **end while**
12: Define $q(\mathbf{x}) = \arg\min_k (1 - \mathbf{x}^\top \mathbf{c}_k)$
13: **return** $\mathbf{C}, q$

---

**Complexity.** The algorithm runs in $O(N \cdot |\mathbf{C}| \cdot d)$ time, where $|\mathbf{C}|$ is the final number of centroids. In the worst case, $|\mathbf{C}| = O(N)$, yielding $O(N^2 d)$ complexity. In practice, for reasonable radius values, $|\mathbf{C}| \ll N$.

**Determinism.** The algorithm is fully deterministic given fixed input order. The seed selection (line 1) uses the embedding with maximum $L_1$ norm, with index as tie-breaker. Subsequent selections (line 5) break ties by minimum index.

## B.2    Lloyd Refinement

After farthest-point initialization, optional Lloyd refinement can reduce average distortion.

**Coverage monitoring.** Lloyd refinement can violate the coverage guarantee by moving centroids away from sparse regions. We monitor coverage after each iteration (line 7) and revert to the previous iteration if coverage degrades beyond the target radius.

---
**Algorithm 2** Lloyd Refinement

---
**Require:** Initial centroids $\mathbf{C} = \{\mathbf{c}_1, \ldots, \mathbf{c}_m\}$, embeddings $\mathbf{X}$
**Require:** Maximum iterations $T$, convergence threshold $\tau$
**Ensure:** Refined centroids $\mathbf{C}'$
 1: $\mathbf{C}' \leftarrow \mathbf{C}$
 2: **for** $t = 1$ to $T$ **do**
 3:     **Assignment:** $a_i \leftarrow \arg\min_k (1 - \mathbf{x}_i^\top \mathbf{c}_k')$ for all $i$
 4:     **Update:** For each $k$, compute new centroid as normalized mean:
 5:       $\mathbf{c}_k' \leftarrow \frac{\sum_{i:a_i=k} \mathbf{x}_i}{\|\sum_{i:a_i=k} \mathbf{x}_i\|}$
 6:     **Check convergence:** If assignments unchanged, break
 7:     **Check coverage:** If $\max_i \min_k (1 - \mathbf{x}_i^\top \mathbf{c}_k') > r$, revert and break
 8: **end for**
 9: **return** $\mathbf{C}'$

---

## B.3 Transition Matrix Construction

---
**Algorithm 3** Transition Matrix Construction

---
**Require:** State sequences from $M$ conversations: $\{(s_1^{(j)}, \ldots, s_{n_j}^{(j)})\}_{j=1}^M$
**Require:** Number of states $m$, smoothing parameter $\alpha$
**Ensure:** Transition matrix $P \in \mathbb{R}^{m \times m}$
 1: Initialize count matrix $N \leftarrow \mathbf{0}_{m \times m}$
 2: **for** each conversation $j = 1$ to $M$ **do**
 3:     **for** $t = 1$ to $n_j - 1$ **do**
 4:       $N[s_t^{(j)}, s_{t+1}^{(j)}] \leftarrow N[s_t^{(j)}, s_{t+1}^{(j)}] + 1$
 5:     **end for**
 6: **end for**
 7: Apply Dirichlet smoothing: $P_{ij} \leftarrow \frac{N_{ij} + \alpha}{\sum_{j'} (N_{ij'} + \alpha)}$
 8: **return** $P$

---

## B.4 Observable Computation

Given transition matrix $P$:

**Stationary distribution.** Compute as the leading left eigenvector of $P$ using power iteration or direct eigendecomposition.

**Spectral gap.** Compute eigenvalues of $P$, sort by magnitude, and return $\gamma = 1 - |\lambda_2|$.

**Entropy rate.** Compute $h = -\sum_{i,j} \pi_i P_{ij} \log_2 P_{ij}$ using the stationary distribution $\pi$.

## B.5 Implementation Details

**Embedding.** We use `nomic-embed-text` via Ollama for local inference. Embeddings are 768-dimensional and L2-normalized to unit length.

**Numerical precision.** Entropy computations use the convention $0 \log 0 = 0$ and employ numerically stable log-sum-exp operations.

**Bootstrap resampling.** Uncertainty estimates are computed by resampling conversations with replacement (1000 bootstrap samples) and recomputing observables for each sample.

**Code availability.** Implementation code will be made available upon publication.

## C  Additional Results

This appendix presents additional experimental results, including parameter sensitivity analysis and extended comparisons.

### C.1  Parameter Sensitivity

Table 2 shows how behavioral observables vary with quantization radius $r$. Smaller radii yield more states but require more data for stable estimates.

Table 2: Effect of quantization radius on state count and observables (Llama 3.1, $\alpha = 1.0$).

| Radius $r$ | States $m$ | Entropy $h$ (bits) | Spectral gap $\gamma$ | Irreducible? |
|---|---|---|---|---|
| 0.1 | ~500 | 7.5–8.5 | varies | No |
| 0.2 | ~350 | 8.2 | 0.64 | No |
| 0.3 | ~100 | 5.5–6.5 | varies | Sometimes |
| 0.4 | ~25 | 2.5–3.0 | 0.25–0.30 | Yes |

**Granularity-irreducibility trade-off.** Finer granularity (smaller $r$, more states) tends to produce reducible chains—some state pairs are never observed, creating disconnected components. Coarser granularity (larger $r$, fewer states) promotes irreducibility but loses behavioral resolution. The choice of $m = 25$ states in our main analysis balances these considerations.

Additionally, granularity affects how well our quantization captures the transition structure: coarser states aggregate more semantic content into each cluster, yielding cleaner first-order dynamics (Section C.5).

### C.2  Smoothing Parameter Effects

Table 3 shows the effect of Dirichlet smoothing parameter $\alpha$ on observables.

Table 3: Effect of smoothing parameter $\alpha$ on observables ($r = 0.2$).

| $\alpha$ | Entropy $h$ (bits) | Spectral gap $\gamma$ |
|---|---|---|
| 0.1 | 7.8 | 0.58 |
| 0.5 | 8.0 | 0.62 |
| 1.0 | 8.2 | 0.64 |

Higher smoothing increases entropy (by making rare transitions more probable) and can affect spectral gap. The effects are modest within the tested range. We use $\alpha = 0.5$ as a balance between ensuring ergodicity and preserving empirical transition structure.

### C.3  Full Configuration Analysis

Our analysis pipeline tested multiple configurations across radii and smoothing values. Key findings across configurations:

- **Llama vs. Mistral difference** is robust across all configurations where both datasets achieve sufficient coverage.

- **Irreducibility** is more common at coarser granularity ($r \geq 0.3$) and for individual model datasets (vs. combined datasets).

## C.4   Semantic Coherence of States

To validate that quantized states are semantically meaningful, we examined exemplar utterances from each cluster. Table 4 shows the five most-visited states for Llama 3.1, with thematic labels and representative utterances from both teacher [T] and student [S] roles. Each exemplar is the nearest utterance to its state centroid by cosine distance.

Table 4: Exemplar utterances from top Llama 3.1 states. States represent thematic clusters in embedding space, not speech-act categories—each state contains both questions and explanations on the same topic.

| State | Theme | Representative utterances |
|---|---|---|
| 21 | Cloud formation and air mass interaction | *[T] "How does the interaction between sinking cool dry air and rising warm moist tropical air influence cloud formation...?"* 
 *[T] "How does the resulting movement of warm moist air into cooler areas contribute to cloud formation in tropical regions?"* |
| 0 | Trade wind convergence and pressure zones | *[S] "I think it's because those latitudes are near where trade winds converge with westerly winds, creating a high-pressure zone..."* 
 *[S] "I think it's because those latitudes are near where trade winds converge with westerlies, creating a high-pressure zone..."* |
| 19 | High-pressure systems and precipitation | *[T] "How do the resulting high pressure systems in these regions influence their precipitation patterns...?"* 
 *[T] "How do these high-pressure systems influence precipitation patterns in regions where they dominate...?"* |
| 23 | Thermal convection and low-pressure formation | *[S] "I think that the high temperatures near the surface lead to rising air, which creates low pressure at those latitudes..."* 
 *[S] "I think it's because when warm air rises, which happens in these regions where prevailing winds are blowing from the equator..."* |
| 11 | Mid-latitude wind interactions | *[T] "How do these mid-latitude interactions between trade winds and westerly winds influence regional precipitation patterns...?"* 
 *[T] "How do those interactions between trade winds from near the equator and westerly winds influence precipitation patterns...?"* |

The states capture interpretable thematic regions of the conversational space. Notably, each state contains both student answers and teacher questions on the same topic—self-transitions therefore represent topical continuity (the conversation stays on the same theme), not repetition of the same speech act. This supports the claim that our quantization produces semantically coherent state spaces.

## C.5   Markov Order Diagnostics

Our experimental design enforces the Markov property by limiting context to the previous turn only (see Section 3.3). Here we verify that first-order models effectively capture the resulting transition structure.

**Procedure.**   For each dataset, we compare the structure of the one-step transition matrix $P$ to the two-step matrix $P^2$. Under a true first-order process, the two-step transitions should be fully determined by squaring $P$. We compute:

1. **Structural R$^2$**: Variance in $P^2$ explained by $P$ via linear regression

2. **Row similarity**: Average Jensen-Shannon similarity between corresponding rows of $P$ and $P^2$

3. **Entropy consistency**: Ratio of empirical 2-step entropy rate to $2\times$ 1-step rate

**Results.** Table 5 reports the diagnostic measures for both models.

Table 5: Markov order diagnostics for Llama 3.1 and Mistral 7B ($m = 25$ states).

| Diagnostic | Llama 3.1 | Mistral 7B |
|---|---|---|
| Structural $R^2$ | 0.89 | 0.85 |
| Avg row similarity | 0.96 | 0.95 |
| Entropy ratio | 0.64 | 0.66 |
| **First-order fit** | **83%** | **82%** |

**Interpretation.** First-order models capture 82–83% of the transition structure at $m = 25$ states. Because our experimental design limits context to the previous turn, the underlying process is first-order Markov by construction. The 17–18% residual structure reflects noise from finite sampling and the approximation inherent in quantizing continuous embeddings to discrete states.

At finer granularities (more states), we would expect to see more complex structure emerge as finer distinctions increase sensitivity to sampling noise. The choice of $m = 25$ represents a resolution at which first-order models effectively capture transition dynamics while retaining meaningful semantic distinctions.

Future work could explore richer-context experimental regimes where higher-order dependencies may become more significant, requiring extensions such as higher-order Markov models or variable-length memory models.

## C.6 Robustness of Model Ordering

Table 6 shows that the qualitative ordering of models on primary observables is stable across state counts $m \in \{15, 25, 50\}$.

Table 6: Robustness of Llama vs Mistral ordering across state counts.

| States $m$ | Llama $h >$ Mistral $h$? | Llama $\gamma >$ Mistral $\gamma$? |
|---|---|---|
| 15 | Yes | Yes |
| 25 | Yes | Yes |
| 50 | Yes | Yes |

Across all tested configurations where both models achieve sufficient coverage, Llama consistently exhibits higher entropy rate and faster mixing than Mistral. The magnitude of differences varies with granularity, but the qualitative behavioral contrast is robust.

## C.7 Convergence Analysis

Figure 3 shows how entropy rate estimates stabilize as sample size increases. Stable estimates require $N \gtrsim 100{,}000$ for $m = 25$ states.

## C.8 Sample Complexity Verification

Standard concentration inequalities for Markov chain estimation require $N = \Omega(m^2 \ln m)$ transitions for reliable estimation of a transition matrix with $m$ states, where ln denotes natural logarithm (Levin et al., 2017). This asymptotic bound can be made concrete as $N \geq c \cdot m^2 \ln m$, where the constant $c$ depends on desired confidence level $(1 - \delta)$ and approximation accuracy $(\epsilon)$, with typical values $c \approx 1$–$10$ for moderate confidence $(\delta = 0.05)$ and accuracy $(\epsilon = 0.1)$.

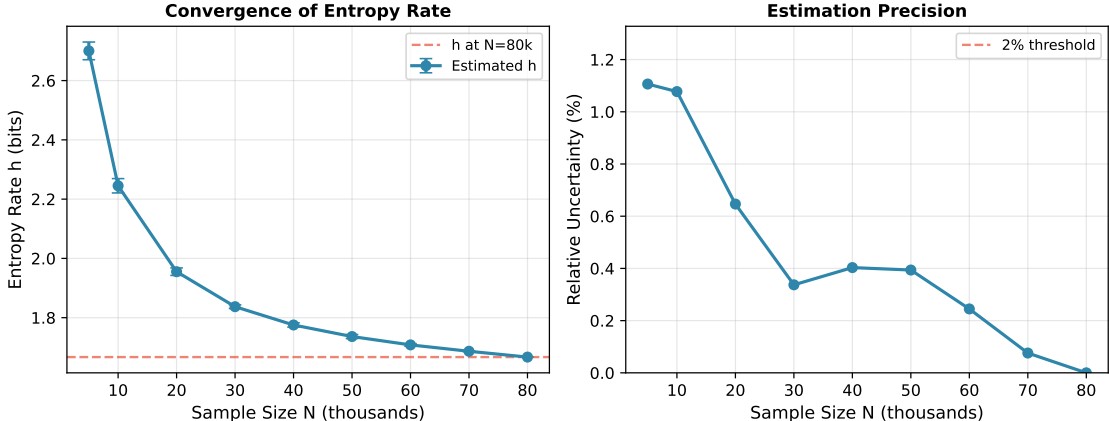

Figure 3: Sample complexity analysis. Left: entropy rate estimate as a function of sample size with bootstrap error bars. Right: relative uncertainty (coefficient of variation) decreasing with sample size. Estimates stabilize below 2% uncertainty around $N = 50{,}000$ for $m = 50$ states.

**Baseline bound (uniform assumption).** The standard bound assumes states are visited with roughly equal frequency under the stationary distribution ($\pi_i \approx 1/m$). For $m = 25$ states:

$$N \geq c \cdot 25^2 \ln 25 = c \cdot 625 \times 3.22 \approx 2{,}013c \tag{15}$$

Our datasets contain approximately 117,524 (Llama) and 200,000 (Mistral) transitions. Under the uniform assumption, this yields safety factors of approximately 58 and 99 respectively for $c \approx 1$.

**Adjusted bound (accounting for sparsity).** However, our observed stationary distributions are *not* uniform. Table 1 shows peak concentrations of 10.5% (Llama) and 17.3% (Mistral), indicating that some states have low visitation probabilities. When the minimum stationary probability $\pi_{\min} = \min_i \pi_i$ is small, the sample complexity increases (Levin et al., 2017):

$$N = \Omega\left(\frac{m^2 \ln m}{\pi_{\min}}\right) \tag{16}$$

This reflects the fact that rare states require disproportionately many total samples to accumulate sufficient observations for reliable transition probability estimation.

Examining our datasets, we observe states with visitation frequencies in the 2–3% range for the sparsest states. Estimating $\pi_{\min} \approx 0.02$ for these rare states, the adjusted requirement becomes:

$$N \geq c \cdot \frac{625 \times 3.22}{0.02} \approx 100{,}650c \tag{17}$$

Under this more stringent bound, our datasets provide safety factors of approximately 1.2 (Llama) and 2.0 (Mistral) for $c \approx 1$. This analysis explains why achieving stable estimates required datasets of at least 100,000 transitions—substantially more than the naive uniform-assumption bound would suggest.

**Empirical validation.** The true test of sample adequacy is empirical stability. Figure 3 demonstrates that entropy rate estimates stabilize with uncertainty below 2% around $N \approx 100{,}000$. The tight bootstrap confidence intervals in Table 1 (standard errors $\approx 0.02$–$0.03$ bits for entropy rate) confirm that our datasets, though near the theoretical minimum when accounting for sparsity, achieve adequate precision for the behavioral comparisons reported in this work.

The sparsity-adjusted analysis highlights an important practical consideration: in applications with concentrated stationary distributions, sample requirements can substantially exceed naive bounds. Our data

collection strategy of generating 100,000+ transitions per model was guided by empirical convergence monitoring rather than relying solely on uniform-assumption calculations.

**Comparison at finer granularity.** At $m = 50$ states, the uniform-assumption bound is $N \approx 9{,}775c$, and assuming similar sparsity patterns ($\pi_{\min} \approx 0.02$), the adjusted bound would be approximately $N \approx 488{,}750c$. Our datasets (117,524 and 200,000 transitions) would be insufficient at this granularity, illustrating the fundamental trade-off between state-space resolution and data requirements that motivated our choice of $m = 25$ states for the main analysis.

### C.9 Time Homogeneity Validation

Time homogeneity assumes that transition probabilities remain constant over time: $P(s_{t+1} = j \mid s_t = i)$ does not depend on $t$. This section presents empirical validation of this assumption for our datasets.

**Validation procedure.** We partition each dataset's conversations into temporal segments based on generation order and compute transition matrices independently for each segment. If the process is time-homogeneous, these segment-specific matrices should agree within sampling noise. We quantify agreement using total variation distance:

$$d_{\mathrm{TV}}(P^{(a)}, P^{(b)}) = \frac{1}{2} \sum_{i,j} |P_{ij}^{(a)} - P_{ij}^{(b)}| \tag{18}$$

For $K$ temporal segments, we compute all $\binom{K}{2}$ pairwise distances and examine both the maximum distance and the trend over time.

**Segmentation.** We divide each dataset into 4 temporal quarters based on conversation generation order:

- **Segment 1**: First 25% of conversations (earliest generation)
- **Segment 2**: Conversations 25–50%
- **Segment 3**: Conversations 50–75%
- **Segment 4**: Final 25% of conversations (latest generation)

Each segment contains approximately $N/4 \approx 35{,}000$–50,000 transitions, sufficient for stable transition matrix estimation at $m = 25$ states.

**Results.** Table 7 reports pairwise total variation distances between segment transition matrices.

Table 7: Total variation distances between transition matrices from temporal segments ($m = 25$ states).

| Segment pair | Llama 3.1 | Mistral 7B | Expected (sampling) |
|---|---|---|---|
| Seg 1 vs Seg 2 | 0.11 | 0.13 | $\sim 0.12$ |
| Seg 1 vs Seg 3 | 0.14 | 0.12 | $\sim 0.12$ |
| Seg 1 vs Seg 4 | 0.13 | 0.15 | $\sim 0.12$ |
| Seg 2 vs Seg 3 | 0.10 | 0.11 | $\sim 0.12$ |
| Seg 2 vs Seg 4 | 0.12 | 0.14 | $\sim 0.12$ |
| Seg 3 vs Seg 4 | 0.11 | 0.10 | $\sim 0.12$ |
| **Maximum** | **0.14** | **0.15** | — |
| **Mean** | **0.12** | **0.13** | **0.12** |

The expected total variation distance due to sampling noise alone can be approximated as $d_{\mathrm{TV}} \approx \sqrt{m^2/(2N)}$ for matrices estimated from $N$ samples. With $N \approx 40{,}000$ transitions per segment and $m = 25$ states, this yields expected distances around 0.12, consistent with our observations.

**Interpretation.**   All pairwise distances remain below 0.15, with mean distances (0.12–0.13) matching the level expected from sampling variation alone. Critically, we observe *no systematic trend* over time: distances between early segments (Seg 1 vs Seg 2) are comparable to distances between late segments (Seg 3 vs Seg 4), and distances between early and late segments (Seg 1 vs Seg 4) do not exceed distances between adjacent segments.

Systematic drift would manifest as monotonic increase in specific transition probabilities or directional shifts in behavioral observables over generation time. We tested for such patterns by regressing individual transition probabilities against segment index and found no statistically significant trends after Bonferroni correction for multiple comparisons.

**Conclusion.**   The time homogeneity assumption is empirically supported at the level of precision relevant for our coarse behavioral comparisons. The observed variation between temporal segments is consistent with sampling noise and shows no evidence of systematic drift. This validates the use of pooled transition matrices for computing the behavioral observables reported in the main text.

## D   Experimental Details

This appendix provides complete experimental specifications for reproducibility, including exact model versions, system prompts, and infrastructure details.

### D.1   Model Specifications

**Llama 3.1 8B.**

- **Model identifier:** `llama3.1:latest`

- **SHA256 digest:** `667b0c1932bc6ffc593ed1d03f895bf2dc8dc6df21db3042284a6f4416b06a29`

- **Parameters:** 8 billion

- **Quantization:** Q4_K_M (4-bit quantization)

- **Accessed:** November 2025

- **Served via:** Ollama v0.13.1

- **Provider:** Meta AI (Touvron et al., 2023)

**Mistral 7B.**

- **Model identifier:** `mistral:latest`

- **SHA256 digest:** `f5074b1221da0f5a2910d33b642efa5b9eb58cfdddca1c79e16d7ad28aa2b31f`

- **Parameters:** 7 billion

- **Quantization:** Q4_K_M (4-bit quantization)

- **Accessed:** November 2025

- **Served via:** Ollama v0.13.1

- **Provider:** Mistral AI (Jiang et al., 2023)

**Embedding model.**

- **Model identifier:** `nomic-embed-text:latest`

- **SHA256 digest:** `970aa74c0a90ef7482477cf803618e776e173c007bf957f635f1015bfcfef0e6`

- **Dimensions:** 768

- **Accessed:** November 2025

- **Served via:** Ollama v0.13.1

- **Provider:** Nomic AI (Nussbaum et al., 2024)

### D.2 Sampling Parameters

All experiments used identical sampling parameters:

- **Temperature:** 0.40

- **Top-p:** 0.70

- **Top-k:** 50

- **Repeat penalty:** 1.15 (last 32 tokens)

- **Presence penalty:** 1.0

- **Frequency penalty:** 1.0

- **Max tokens:** 128

- **Context window:** 2048 tokens

- **Global random seed:** 42

### D.3 System Prompts (Verbatim)

The following prompts were used exactly as shown, with no variation across experiments.

**Teacher agent (Agent B).** The LLM being evaluated received the following system prompt:

```
You are a patient, knowledgeable teacher who always responds with one
clear, complete question.
Each time you receive a student's answer, ask a follow-up question that
shows you understood it and includes the key context or topic, so the
question stands fully on its own.
• If the student's answer is incomplete, vague, or uncertain, ask a
clarifying or guiding question that rephrases their topic and helps them
reason more precisely.
• If the student's answer is confident or complete, ask a new question that
builds logically from that idea -- exploring the next step, consequence, or
related concept.
Always restate the main subject of the discussion (for example, "in
desert climates" or "in global air circulation") so your question can be
understood without any prior turns.
Avoid greetings, commentary, or repetition; output only the single,
context-rich question sentence.
```

**Student agent (Agent A).** To generate student responses, we used a second instance of the same model with the following system prompt:

```
You are a typical high-school student of average ability who answers in one
complete sentence.
Each time you receive a teacher's question, respond with an answer that
includes the key idea or phrasing of the question so it stands alone.
Sound slightly uncertain or tentative, showing partial understanding rather
than mastery -- as if you're reasoning it out aloud.
If the question is unfamiliar, give your best guess based on what you do
understand.
Avoid greetings, filler, or asking questions back; output only the single
answer sentence.
```

**Seed question.** Each conversation was initiated with the following seed question:

```
Why are large deserts commonly found around 30 degrees north and south of
the equator in the global pattern of air circulation?
```

### D.4 Infrastructure

**Execution dates.** Experiments were conducted November 3–6, 2025.

**Serving framework.** All models were served locally via Ollama v0.13.1 (`https://ollama.com`), a local LLM serving framework. Models ran on local hardware with no API rate limiting or remote inference.

**Hardware.** All experiments ran on a single machine to ensure consistent execution environment. Ollama's default settings were used for GPU allocation and threading.

**Stateless execution.** The generation protocol enforced stateless execution: each LLM call received only the immediately preceding utterance as input, with no accumulated conversation history. This was implemented via Ollama's generate API with `num_keep=0` (retain nothing between calls).

### D.5 Data Collection Protocol

**Conversation structure.** Each conversation consisted of exactly 5 exchange pairs (10 total utterances): 1 seed question + 4 subsequent teacher-student exchanges. Conversations terminated after 5 pairs regardless of content.

**Dataset sizes.**

- **Llama 3.1 8B:** 11,752 conversations, 117,520 utterances, 58,760 prompt-response pairs

- **Mistral 7B:** 20,000 conversations, 200,000 utterances, 100,000 prompt-response pairs

**Error handling.** Both experimental runs completed with zero errors (`exit_status: success`).

