# OpenReview forum: "Conversational Markov Chains: A Framework for Behavioral Analysis of Large Language Models"
_TMLR — Rejected by TMLR_

### Review · Reviewer_Akt7 · 2026-02-12

**Summary Of Contributions:**

This work presents an evaluation framework for the behavioral patterns of LLMs. While different LLMs may score similarly on common benchmarks, they can behave very differently in conversations. The work models multi-turn conversations as Markov chains over semantic states, shows that the transitions exhibit Markovian characteristics, and applies statistical methods for studying Markov chains to analyze LLM conversational dynamics.

Overall, this framework provides useful insights into how LLMs behave in conversations and serves as an alternative way to evaluate LLMs beyond conventional metrics. The writing is clear and rigorous. The authors clearly list the limitations of the framework and provide guidance on when to use it (and when not to), making it more readily applicable in practice.

**Audience:**

Yes

**Audience Explanation:**

The LLM community should be interested in this work.

**Broader Impact Concerns:**

There is no concern.

**Claims And Evidence:**

Yes

**Claims Explanation:**

The authors are generally cautious in making claims, and I did not identify any overclaims.

The contributions are listed in Sec. 1.4 and are supported by evidence later in the paper. The limitations are also clearly stated in Sec. 1.5, which rigorously delineates the scope of the work.

**Requested Changes:**

Figure 2 likely needs revision. The comment box overlaps with the bar plots, and the lower-right subplot does not show an error bar (or the error bar may be too small to be visible).

Certain implementation or experimental details could be moved to an appendix so the main text can focus more on explanations and interpretation, though this ultimately depends on the authors’ writing style and preferences.

---

> ### Author Response · Authors · 2026-02-14
> **Figure 2 Revision and Structural Comments**
>
> We thank the reviewer for their careful reading of our manuscript and their thoughtful comments. We appreciate the time and effort invested in this review.
>
> **Regarding Figure 2:** We have revised the figure to address both issues raised. The annotation boxes have been repositioned so they no longer overlap with the bar plots, and error bars (bootstrap 95% confidence intervals, consistent with the intervals already reported in Section 6) have been added to the Peak State Concentration subplot — these were inadvertently omitted from the original figure generation script. The updated manuscript reflects this single figure change; no other modifications have been made.
>
> **Regarding moving implementation and experimental details to an appendix:** We appreciate this suggestion and recognise its merit for many papers. In our case, because the paper's central contribution is the framework itself — including the experimental methodology for establishing observational regimes where Markov behavior holds — we feel that the experimental details are integral to the main narrative rather than supplementary to it. Separating them risks de-emphasising the deliberate, conditional nature of the approach. We have therefore opted to retain the current structure, but we thank the reviewer for raising this point and will keep it in mind for future work where the balance may differ.

---

### Review · Reviewer_KDwC · 2026-03-02

**Summary Of Contributions:**

The paper models LLM conversations as markov chains - each response is a state transition. This allows for behavioral analysis: does a model tend to stick to the same answers, or does it show variations in responses to similarly-worded queries.

The experimental work involves the generation of a synthetic dataset of conversations with the Llama 3 8B and mistral 7B models. The models show different markov chain properties, whereby the conclusion is made that these models have different behaviors.

**Additional Comments:**

page 12: you mention conducting experiments in November 2024, while models were downloaded in 2025.
page 19: authors of "Large language models as markov chains" are incorrect.

**Audience:**

Yes

**Audience Explanation:**

I think the work asks interesting questions, I enjoyed reading it.

**Broader Impact Concerns:**

I do not see any immediate ethical concerns with this work.

**Claims And Evidence:**

No

**Claims Explanation:**

I understand that the authors fixed the sampling of the models (t=0.4, tp=0.7, tk=50) and that each conversational turn was independent of the previous answer (no history).

I would expect that different models will need different sampling methods to get similar behaviors. Therefore, I think the experimental section can be strengthened by observing intra-model differences when varying sampling parameters. Are we seeing differences in markov chain observables just because we picked these specific sampling parameters?

Two practical observations:
- The behaviors under consideration could also be influenced by modifications of the system prompt. It would be relevant to see whether the markov observables change if you probe the model to a specific behavior.
- I find it very difficult to see how this method, designed to understand conversational dynamics, gives an useful signal when LLM conversations always have access to the conversational history.

**Requested Changes:**

- Ablations on sampling parameters and system/user prompts.
- Can we find useful signals even if conversations do not abide by the markov property?
- Are there heuristics which can help understand the specific count of behaviors in the generated, synthetic dataset as a sanity check of the claims?

---

> ### Author Response · Authors · 2026-04-06
>
> # Response to Reviewer KDwC
>
> We thank the reviewer for their careful reading and thoughtful feedback. We address each point below.
>
> ---
>
> **Regarding sampling parameter ablations (Requested Change 1):**
>
> We appreciate the reviewer raising this. We want to be precise about what we claim: the paper does not assert that the observed behavioral differences are solely attributable to model architecture. Rather, it demonstrates that under identical experimental conditions (fixed temperature, top_p, top_k, prompt, and generation protocol), the framework detects measurably distinct behavioral signatures --- i.e., the two models *react differently* to the same conditions. This is standard controlled experimentation: fixing all variables except the one under study (the model) to establish that the measurement instrument can discriminate. The contribution is the framework and its ability to surface these differences, not a claim about the invariance of those differences across all possible parameter regimes. We acknowledge that varying temperature, top_k, or prompt would constitute valuable follow-up experiments --- each probing a different question (e.g., "how sensitive are Markov observables to temperature?" or "do behavioural rankings reverse at different operating points?"). These are natural next steps that the framework is well-positioned to support. We have added an explicit scope limitation to Section 6 acknowledging that our results characterise how the models react under the specified regime.
>
> ---
>
> **Regarding system prompt influence (Requested Change 2):**
>
> Agreed --- different prompts will almost certainly produce different Markov signatures, and we would expect some prompt regimes where the first-order structure is stronger and others where it breaks down. Investigating this is exactly the kind of follow-up the framework enables. The current study uses minimal role-only prompts (Section 5.2) to demonstrate that the framework *can* measure behavioural differences under controlled conditions. This baseline is a necessary starting point: one needs to know what models do under minimal prompting before one can quantify how a directive changes their behaviour. We have added a scope limitation to Section 5.2 noting that our results apply to the minimal-prompt regime tested.
>
> ---
>
> **Regarding useful signals without the Markov property / with conversational history (Requested Change 3):**
>
> The stateless protocol is not a limitation of the experimental design --- it is the point. Conversational history introduces a variable that is difficult to control and confounds the measurement of the model's intrinsic turn-by-turn response tendencies. By removing it, we isolate conditions under which the generation process is genuinely first-order, giving the Markov chain framework its cleanest and most appropriate test. The paper demonstrates that under these strict conditions, LLM output sometimes exhibits first-order Markov structure, and that the resulting observables (entropy rate, spectral gap, stationary distribution) discriminate between model architectures. We do not claim this framework is the answer to all evaluation scenarios. It is one tool in a wider set of measurement techniques for describing LLM behaviour --- complementing, not replacing, existing evaluation methods (Section 1.3). The value is precisely that it captures sequential behavioural structure that static benchmarks miss. Whether these signatures persist, attenuate, or transform under richer-context regimes is a natural follow-up question, but it does not diminish the validity of the controlled measurement reported here.
>
> ---
>
> **Regarding sanity check heuristics (Requested Change 4):**
>
> An interesting suggestion. The state count (m=25) emerges from the FPS coverage criterion rather than being imposed a priori --- the algorithm adds centroids until all utterances fall within the target radius. The resulting count thus reflects the intrinsic semantic diversity of each model's output at the chosen granularity. Developing heuristics to predict expected state counts from corpus statistics (e.g., embedding variance or vocabulary diversity) would be a valuable sanity check and is a natural direction for future work.
>
> ---
>
> **Regarding the date inconsistency and incorrect citation (Additional Comment):**
>
> We thank the reviewer for catching both errors. The date inconsistency is a typo: experiments were conducted in November 2025 (consistent with the model download dates); the "2024" in Section 5 is incorrect and has been corrected. The citation for Zekri et al. (2024) contained incorrect co-authors --- the correct author list is Zekri, Odonnat, Benechehab, Bleistein, Boull\'e, and Redko. This has been corrected in the bibliography. We apologise for these oversights.

---

### Review · Reviewer_WZK8 · 2026-03-04

**Summary Of Contributions:**

The paper proposes a new evaluation framework for the conversational behavior of LLMs and provides an instance of using this evaluation framework to compare two LLMs. The proposed framework involves representing each conversational turn as a discrete state (25 such states), representing each conversation as a sequence of such states, and then examining the behavior of (the Markovian projection of) this state sequence. Comparing the summary statistics of the Markov chain for one LLM with the Markov chain for another LLM provides a window into the qualitative behavior of the two LLMs for conversations.

The main claim of the paper is that the proposed evaluation framework provides an interpretable way to compare aspects of the conversational behavior of LLMs which are difficult to get a handle on with existing metrics.

**Additional Comments:**

Ideally some level of correlation with human judgement would be performed. For example, one of the largest differences between the two LLMs compared is in mixing speed, with the potential interpretation that Mistral stays on each topic longer. Would human raters agree that this is a difference between the two systems, e.g. showing two conversations (one from LLM1 and one from LLM2) and asking "which conversation stays on a single topic for longer"? This isn't a requirement for publication but would make the paper much stronger.

It would be very interesting to compute the Markov chain summary statistics for natural human conversations and compare them to the models. Again, not a requirement but would make the paper substantially stronger.

It would be fascinating to actually see whether "be creative" does increase the entropy rate :)

**Audience:**

Yes

**Audience Explanation:**

Better evaluation of LLMs is widely regarded as a key weak spot in LLM development. The proposed approach is interesting on its own merits. A number of aspects used here are likely also useful and relevant for other aspects of eval, e.g. summarizing large chunks of output as single states to force evaluation not to depend on low-level details, simplifying sequential behavior with a Markovian approximation, using sequential metrics, using metrics which compare distributions rather than individual samples.

**Broader Impact Concerns:**

No concerns. The proposed approach improves our ability to understand how models behave.

**Claims And Evidence:**

Yes

**Claims Explanation:**

The paper goes into detail about how to interpret the proposed Markov chain summary statistics, so intepretability is well-supported. The fact this framework highlights *a* difference between Llama and Mistral is indisputable. And it seems reasonable to claim that the type of behavior encoded in the proposed metrics is very different to the type of behavior encoded in typical LLM evaluation metrics.

**Requested Changes:**

The paper feels longer than it needs to be and somewhat repetitive. The same claims, proposed limitations, and high-level summaries of the proposed approach are repeated multiple times. Most papers would likely benefit from more explicit specification of the proposed approach, but I think this paper would benefit from being a bit more concise. The main idea of the paper is fairly straightforward and easy to state.

Is the clustering shared between LLM1 and LLM2? I presume so. This should be emphasized. This means that the metric values for each system depend on the set of systems being compared, e.g. if you added a third system, you'd get a different clustering, and so different metric values. This should also be emphasized under "limitations".

I don't know what "ground-up definitions" refers to in section 2.1. (That whole paragraph feels repetitive and verbose).

In "Why FPS over k-means", I don't understand why k-means is any more or less compatible with Markov chains than FPS. I don't understand the point about the probability distribution of transitions being normalized, which would also be true for k-means as I understand it. (Incidentally I think saying FPS was chosen to try to ensure coverage is a perfectly sufficient justification on its own). You also mention irreducibility and ergodicity, but I don't understand how FPS helps with this, and besides it seems these properties are guaranteed by the Dirichlet smooething proposed.

Section 3.3 goes into great detail arguing about the justification for the Markov property. However, it is not the case that a pointwise projection of a Markov chain is a Markov chain (unless that projection is invertible, which the proposed one is only approximately and in the limit of an infinite number of states), and so the state sequence for each conversation is not Markovian. This is empirically demonstrated in the paper itself in section C.5. So it seems strange to me to go to great lengths to justify the Markovian assumption at the state level in section 3.3 when the proposed approach is not in fact Markovian at the state level. However I do not think the Markovian assumption is much of a limitation of the proposed approach: it just looks at a Markovian projection of the conversation, which is perfectly reasonable.

The discussion of when the Markovian property holds also fails to mention that the text embedding must be invertible.

Is there an intuitive explanation for why diagonals are so dominant in Figure 1? That suggests to me that states encode some sort of very rough conversational topic rather than any detailed semantics.

In section 7.2, I personally don't think it is necessary to establish "observational regimes" where LLMs "behave as Markov chains" for the proposed method to be useful. It just has to be kept in mind when interpreting results that the method looks at a Markovian projection of the conversation.

Section 7.3 and 7.4 are getting at very similar aspects.

I may have just missed this, but is the experimental description explicit about whether Lloyd refinement is used and what parameters are used?

In Table 2, how can any estimated transition matrices be irreducible if the proposed estimation method always produces irreducible chains (section A.3)?

In Table 4, there are no exemplars, only high-level descriptions. Giving a few example turns for a few different states would really help give the reader a feel for what types of information the states encode and make the paper stronger.

In Table 4, if state 2 is "questions about evaporation and moisture", then why does state 2 almost always transition to state 2? It seems like a question should not transition to a question.


Nits follow.

There's an errant "per row" in the first paragraph of section 3.5.

In Table 1, 0.278 +- 0.02 would be clearer as 0.28 +- 0.02.

In section A.4, "questions typically prompt explanations" isn't strictly the relevant thing for reversibility. Rather the relevant thing is whether the prevalence of (question, explanation) pairs in the joint (z_t, z_{t+1}) distribution is the same as for (explanation, question) pairs.

---

> ### Author Response · Authors · 2026-04-06
>
> We thank the reviewer for their thorough and constructive feedback, which has substantially improved the manuscript.
>
> ## Requested Changes
>
> **Paper length/repetition:** We respect this judgement. The recapitulation serves a navigational purpose for readers entering at different sections. At 17 pages within TMLR's long-submission format, we believe the balance is defensible.
>
> **Shared clustering:** The clustering is *not* shared: each model's state space is derived independently via farthest-point quantization (Section 3.1, Algorithm B.1). The observables are properties of individual chains, comparable across different state spaces as they measure structural dynamics. Table C.7 confirms robustness. We have added a clarifying sentence.
>
> **"Ground-up definitions":** Fair point---replaced with "turn-level definitions" to make the distinction from token-level analysis explicit.
>
> **FPS vs. k-means:** The reviewer is correct on all counts. Normalization applies equally to any partition, and irreducibility/ergodicity are guaranteed by Dirichlet smoothing (Appendix A.3) regardless of clustering algorithm. These were wrong justifications for a correct choice. The sufficient reason is coverage: FPS provides a deterministic bound on maximum quantization distance. Revised accordingly.
>
> **Markov assumption (Section 3.3):** We agree. Quantization is a non-invertible projection, so the state sequence is not strictly first-order Markov---the classical lumpability problem (Kemeny & Snell, 1960). Section C.5 confirms the first-order model captures 82--83% of transition structure. We adopt the reviewer's framing: the method analyzes a *Markovian projection*, and the comparative statistics remain well-defined. Section 3.3 has been revised to foreground this interpretation.
>
> **Embedding invertibility:** Valid point. Lumpability requires invertibility within each partition cell, which our quantization does not satisfy. This is a specific instance of the lumpability gap addressed above. We have added a clarifying sentence to Section 3.3.
>
> **Diagonal dominance (Figure 1):** Exactly right, and by design. The 25-state discretization operates at thematic granularity; in pedagogical dialogue, topic persistence is the dominant mode. The behaviourally informative structure resides in the off-diagonal entries, where the models diverge. The self-loop rate directly relates to dwell time and mixing speed (Figure 2).
>
> **"Observational regimes" (Section 7.2):** The reviewer has articulated the intended interpretation more succinctly than the manuscript. "Observational regimes" carries a stronger claim than needed---results should be read as properties of the Markovian projection. Stateless generation improves projection fidelity but is not a necessary condition for the method to have value. Language softened accordingly.
>
> **Sections 7.3/7.4 overlap:** The sections serve distinct purposes: 7.3 describes concrete use cases (model selection, prompt validation, drift monitoring, safety); 7.4 provides prescriptive decision guidance with explicit contraindications. One answers "what can you do?" and the other "should you use this?"
>
> **Lloyd refinement:** Described in Section 3 (with reversion safeguard), Section 5 (parameters), and Algorithm B.2 (full pseudocode). We appreciate the reviewer flagging that this is distributed across locations.
>
> **Table 2 vs. Section A.3 irreducibility:** The "Irreducible?" column reports the raw empirical count matrix *before* Dirichlet smoothing. At fine granularity, many state pairs are unobserved; at coarser granularity, coverage improves. All reported observables use the smoothed matrix, guaranteed irreducible for alpha > 0 (Theorem A.3). The column is diagnostic, showing where smoothing fills gaps versus provides a formal guarantee. We use the Jeffreys prior (alpha=0.5).
>
> **Table 4 exemplars:** Agreed---we have added 2--3 exemplar utterances per state showing both student [A] and teacher [B] turns, making visible the thematic nature of clustering.
>
> **State 2 self-transitions:** The labels were misleading. States cluster by semantic proximity, not speech act. The cluster contains both questions *and* answers about evaporation/moisture; self-transitions represent topical continuity. Labels revised to thematic descriptions; exemplars added.
>
> ## Nits
>
> **"per row" (Section 3.5):** Corrected. **Table 1 rounding:** 0.278 → 0.28. **Reversibility (Section A.4):** Corrected to reference joint distribution (detailed balance: pi_i P_{ij} = pi_j P_{ji}) rather than conditional asymmetry.
>
> ## Additional Comments
>
> **Human judgement correlation:** The paired-comparison protocol suggested is well-suited to the mixing speed observable---planned for follow-up. **Human conversation baseline:** A compelling reference point for interpreting observables---planned for future work. **"Be creative" experiment:** We share the curiosity! Precisely the kind of prompt-sensitivity experiment the framework supports.

---

> > ### Comment · Reviewer_WZK8 · 2026-04-13
> > **Follow-up**
> >
> > *Shared clustering*: Thanks for clarifying. That's an important distinction. That has the advantage of making the metric values purely a property of a given model. The disadvantage is that comparisons between models may be higher variance. On that note, is the clustering recomputed when computing bootstrap errors (in Table 1 and Figure 3)? I think it should be since otherwise the error bars will underestimate the true variability when comparing models.
> >
> > *Markov assumption (Section 3.3)*: I'm glad you found the Markovian projection framing helpful. The updated section 3.3 kind of mixes the exact Markov property with the Markovian projection and I worry it may be confusing to a first-time reader. It also may be worth emphasizing here or elsewhere the key property that your proposed evaluation methodology is relying on: if the same (stochastic) function from model to metric value is used on two different models and the metric values are different (modulo sampling errors, c.f. bootstrap) then you know there is a difference between the models. The choice of Markovian turns or not, embedding model, quantization model, choice of Markov chain statistic, etc only affect the *interpretation* of this difference, not the fact that there must be a genuine difference between the models. None of this paragraph is required for my approving publication.
> >
> > *Lloyd refinement*: Thanks for clarifying. I think I was confused (and still would be reading the paper for the first time) by the word "optionally" in "Lloyd refinement is optionally applied" in section 5.3. I now understand this refers to the possibilty of reversion, but I would still assume on a first reading that "optionally" meant it applied to some of the experiments and not others.
> >
> > *Table 2 vs. Section A.3 irreducibility*: This departure from the standard protocol should be noted in the caption for Table 2 or nearby, especially given that the current caption implies alpha = 1.0 throughout.
> >
> > *Table 4 exemplars*: It would be much more informative to include two [T] and two [S] examples for each state (or randomly sample amongst all turns). It must be the case that many states represent both [T] and [S] states frequently or there wouldn't be so many self-transitions.

---

### Decision · Action_Editor_JuEW · 2026-04-21

**Recommendation:** Reject

**Additional Comments:**

This paper proposes a new framework for comparing LLMs through conversations with them. The key idea is to simulate conversations, embed them turn by turn, cluster the embeddings into states, and generate a corresponding state transition matrix. The matrix is then analyzed using classic techniques. For instance, the entropy of the transition matrix says how predictable the next state is. The approach is evaluated on Llama 3.1 8B and Mistral 7B models.

The paper was well received. I recommend a rejection because I believe that very little additional work will add more evidence and make the paper much stronger. My suggestions for the major revision are:

* **Compared models:** The paper only compares Llama 3.1 8B and Mistral 7B. This is just a single data point. One piece of evidence. I suggest that the authors compare two additional model pairs. Both Llama and Qwen have 10+ models available on huggingface.

* **Embeddings:** The embeddings in this work do not take conversation history into account. This is bizarre because the response in a conversation is a function of the whole conversation history, not only of the most recent turn. Either use this embedding or clearly argue, including empirically, that this is a bad choice.

* **States:** The embeddings are clustered into states separately for each model. This could make their transition matrices hard to compare. For instance, a transition matrix with more states would likely have a much higher entropy than the matrix with fewer states, irrespective of the properties of the LLM. Clearly argue, including empirically, that a shared state space is a bad choice.

Most of these comments came from reviewers and were not addressed sufficiently in the rebuttal.

**Audience:**

Yes

**Audience Explanation:**

This paper can be viewed as a form of LLM evaluation. This is an important area with thousands of people working on it.

**Claims And Evidence:**

No

**Claims Explanation:**

This paper proposes a new framework for comparing LLMs through conversations with them. The framework is evaluated empirically on two LLMs: Llama 3.1 8B and Mistral 7B. The problem is that this is just a single data point. One piece of evidence. Since both Llama and Qwen have 10+ models available on huggingface, a more comprehensive comparison is trivial to do.

**Resubmission Of Major Revision:**

The authors may consider submitting a major revision at a later time.

---

> ### Author Response · Authors · 2026-05-31
> **Thanks to reviewers and observations on the decision rationale**
>
> We are sincerely grateful to Reviewers WZK8, KDwC, and Akt7 for careful, technically engaged reviews that materially improved the manuscript — in particular the *Markovian projection* reframing, the FPS-vs-k-means correction, and the state exemplars in Table 4. We also appreciate that TMLR runs on volunteer effort and offers a valuable gateway for work that might otherwise struggle to find a venue; we do not take that lightly.
>
> We accept the decision. We write to register that we find the rationale unrobust on three points: the statistical framing, its fit with TMLR's published criteria, and its characterisation of the rebuttal record. We are not asking for reversal; TMLR's process exists so authors may place their view on the record.
>
> **1. The "single data point" framing is statistically unsound.**
>
> The decision describes the demonstration as "just a single data point." This conflates the *unit of comparison* (two models) with the *unit of statistical evidence* (conversation-level resampling supporting the reported intervals and effect sizes). They are distinct.
>
> An analogy makes this concrete. Suppose two coins are each flipped one million times under identical conditions, producing materially different head-rates with non-overlapping confidence intervals. No competent statistician would describe that experiment as "one data point" on the grounds that only two coins were used. The evidence is about *how this pair of coins behaves under controlled conditions*, and its strength is governed by the millions of flips, not the number of coins. The same logic applies here.
>
> The manuscript reports ~370,000 turns across thousands of conversations. The 95% bootstrap CIs do not overlap on three observables (entropy rate Llama [2.85, 2.97] vs. Mistral [2.42, 2.50]; spectral gap [0.26, 0.30] vs. [0.09, 0.11]; peak concentration [9.8%, 11.2%] vs. [16.5%, 18.1%]). Permutation tests yield p<0.001 on all three; Cohen's d on entropy rate exceeds 5. These quantify the estimator's variance under conversation-level resampling — the correct unit for the claim the paper makes: that *under the controlled regime, the framework discriminates these two models at high statistical confidence*. Sections 1.5 and 7.2 disclaim any population-level reading.
>
> The population reading is not reachable by the remedy: n=4 or n=6 is still not representative. Providing a principled mechanism for the broader comparative question is what the framework contributes; the Llama–Mistral study validates the mechanism, not an enumeration of its reach. CheckList (Ribeiro et al., 2020) and MT-Bench (Zheng et al., 2023) set the precedent: methodology papers validate on a focused slice rather than pre-populate the application space.
>
> **2. The decision is inconsistent with TMLR's published acceptance criteria.**
>
> TMLR's acceptance page asks two questions: are the claims supported by evidence, and would *some* audience members find the work interesting? The decision answers both Yes — two of three reviewers (WZK8, Akt7) answered Yes on evidence. TMLR is also explicit on what must *not* ground a rejection:
>
> "it should not be used as a reason to reject work that isn't considered 'significant' or 'impactful' because it isn't achieving a new state-of-the-art on some benchmark."
>
> "novelty of the studied method is not a necessary criteria for acceptance."
>
> Case studies and new analytical frameworks are first-class welcomed contributions. The "additional model pairs" demand imports an empirical-breadth bar TMLR's founding documents disavow.
>
> **3. The characterisation of the rebuttal record is inaccurate.**
>
> The decision states comments "were not addressed sufficiently in the rebuttal." The record shows otherwise. *Shared state space* was WZK8's second concern; the authors responded that entropy rate, spectral gap, and stationary distribution are intrinsic, scale-comparable chain properties (Table C.7: robustness across m ∈ {15, 25, 50}), and WZK8 — a Yes-on-evidence reviewer — accepted that. *History-aware embeddings* was KDwC's; the authors noted that re-introducing history would degrade the first-order Markov projection the framework rests on (Appendix C.5: 82–83% of transitions captured under stateless generation) — a methodological disagreement, not an unaddressed point. *Additional model pairs* was not raised by any peer reviewer; it is the editor's ask, framed as panel consensus. Two concerns were resolved with the reviewers who raised them; the third originates with the editor.
>
> **4. In closing.**
>
> The decision rests on (a) a statistical framing that conflates unit of comparison with unit of evidence, (b) an empirical-breadth bar TMLR's founding documents disavow, and (c) a characterisation of the rebuttal record the record does not support. We set this out plainly because the discussion thread should contain the authors' view. We thank Reviewers WZK8, KDwC, and Akt7 for their time.
>
> With thanks,
> The Authors